# Critical Review of Polymeric Building Envelope Materials: Degradation, Durability and Service Life Prediction

**Marzieh Riahinezhad \***, **Madeleine Hallman and J-F. Masson**

Construction Research Centre, National Research Council of Canada, Ottawa, ON K1A 0R6, Canada; mthallman@uwaterloo.ca (M.H.); massonjfmj@videotron.ca (J.-F.M.)
**\*** Correspondence: Marzieh.Riahinezhad@nrc-cnrc.gc.ca

**Abstract:** This paper provides a critical review of the degradation, durability and service life prediction (SLP) of polymeric building envelope materials (BEMs), namely, claddings, air/vapour barriers, insulations, sealants, gaskets and fenestration. The rate of material deterioration and properties determine the usefulness of a product; therefore, knowledge of the significant degradation mechanisms in play for BEMs is key to the design of proper SLP methods. SLP seeks to estimate the life expectancy of a material/component exposed to in-service conditions. This topic is especially important with respect to the potential impacts of climate change. The surrounding environment of a building dictates the degradation mechanisms in play, and as climate change progresses, material aging conditions become more unpredictable. This can result in unexpected changes and/or damages to BEMs, and shorter than expected SL. The development of more comprehensive SLP methods is economically and environmentally sound, and it will provide more confidence, comfort and safety to all building users. The goal of this paper is to review the existing literature in order to identify the knowledge gaps and provide suggestions to address these gaps in light of the rapidly evolving climate.

**Keywords:** durability; degradation; service life prediction; polymers; building envelope



## 1. Introduction

The building envelope is an essential component of any building. It contains both the structural and non-structural components designed to protect the building occupants from the vagaries of weather. The building envelope is a multi-layered passive element of construction, which, if designed properly, can improve the comfort of the residents and can have a significant effect on the energy efficiency of the building [1–3]. The building envelope should protect the building and its contents from climatic loads, maintain good performance and allow for proper ventilation [4]. The design of the building envelope is unique to the surrounding environment as the geographic location, weather patterns and severity of seasons will affect the performance requirements of the components and the degradation of the building envelope materials (BEMs) [1].

Figure 1 demonstrates the generic structure of a building envelope for dwellings. The primary components of interest in this review are the polymeric materials. From the exterior to the interior, they are the cladding, air barrier/water sheathing membrane (may also be referred to as a weather resistive barrier), insulation and vapour barrier. Sealants, gaskets and fenestration materials in different parts of the building are also covered in this review but they are not shown in this Figure. Some of the components in Figure 1 may be placed in different positions or removed entirely, based on the requirements of local building codes, geographic location and/or building type. For example, in wood frame construction, the exterior sheathing, generally a wood panel, can be the designated air barrier, but it may also be the vapour barrier that is the designated air barrier.

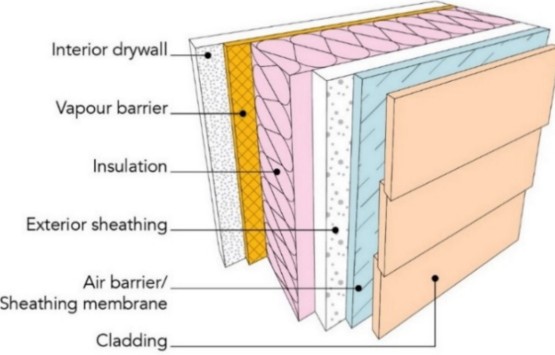

**Figure 1.** Typical multi-layer building envelope.

Each building envelope component has a specific function. The cladding provides protection from wind and water, either in the form of rain, hail or snow, or some level of insulation and protection for the inner layers of the building envelope [5]. The primary purpose of the vapour barrier and air barrier is to prevent diffusion of water vapour and air through the wall, respectively [6]. The insulation prevents heat transfer through the wall. The effectiveness of the insulation will have the biggest impact on energy efficiency and is at times dependent on the effectiveness of the vapour and air barrier [5,6]. Polymers in fenestration and gaskets decrease heat loss and seals the seams of the windows in the building envelope. Sealants improve the overall air quality by sealing any cracks or seams in the envelope and they are often classified as a part of the air barrier system. For this paper, wood materials and paper products have been defined as biopolymers but for the sake of simplicity, their durability has only been examined in key cases, such as wood-polymer composites used in cladding and cellulose insulation.

The building market continues to seek more durable, energy efficient and environmentally friendly building products that remain cost effective. Polymeric materials can provide for a myriad of benefits to the overall function of the building envelope while meeting these needs [7]. Polymers have unique properties, including flexibility, corrosion resistance, low density and cost effectiveness. Consequently, they are increasingly integrated into the building envelope, primarily as non-structural components [1,8]. Since the integration of these materials began, construction and civil engineering have become some of the largest users of polymers of any industry [9]. One of the most useful benefits of polymeric materials is their relatively high thermal resistance, which makes them great insulators when they are foamed. Energy use in commercial buildings can be decreased from 20 to 30% with insulations [10].

Despite the advantages that polymeric products bring to the construction sector, there are several challenges to their use. Polymeric materials have relatively high embodied energy (the energy associated with material production) and carbon footprint compared to other construction materials such as stone, brick or concrete, although it is lower than aluminium [9,11]. Other challenges include poor recyclability and often unknown and untested degradation mechanisms, which lead to uncertain durability and SL expectations [8,12]. The assumed life expectancy of some polymeric materials is listed in Table 1. These life spans are not based on any specific service conditions, but rather general guidelines. Hence, they cannot serve for SLP of polymeric products in the building envelope.

**Table 1.** Assumed lifespan of some plastics; Adapted from Ref. [13].

| Type | Assumed Lifespan (Years) | Building Components |
|---|---|---|
| Butyl Rubber | 2 to less than 35 | Gaskets/Sealants |
| Chloroprene | 2 to less than 40 | Gaskets/Sealants |
| Ethylene Propylene Rubber | Less than 30 | Gaskets |
| Polyethylene | 2 to 15 | Air/Vapour Barriers, Cladding |
| Polyisobutylene | 11 to less than 40 | Sealants |
| Polypropylene | 3 to less than 10 | Air/Vapour Barriers, Cladding |
| Polysulfide | 22 to less than 50 years | Sealants |
| Polyurethane | 7 to 10 | Cladding, Air/Vapour Barrier, Insulations, Sealants |
| Polyvinyl Chloride | 8 to less than 30 | Cladding |
| Silicone | 14 to less than 50 | Air/Vapour Barrier, Sealants |

When a part of the building envelope is damaged or degraded and it can no longer function as intended, there may be a serious impact on the effectiveness of the envelope and it is often difficult and costly to repair or replace. Potential health and safety risks are also present. For example, when the vapour or air barrier fails, moisture enters into the wall components, which can reduce the effectiveness of the insulation, encourage mould growth and in more severe cases, cause damage to the structural components [6]. If the envelope cannot be repaired, the structure may need to be demolished prematurely, which in case of non-recyclable materials, would put a strain on the already expanding landfills [1].

To mitigate risks associated with the use of any material, and premature failure in particular, it is beneficial to assess the SL. SLP methods for polymeric materials are underutilised and, in some cases, underdeveloped. The expansion of SLP methods would produce information regarding life expectancy of materials that would allow for the advantages of polymeric materials to be utilised, while reducing the risks of unexpected failure, damage [12]. There are several SLP methods available [14–19], but there are no comprehensive methods, specifically for polymeric materials used in the building envelope. Durability and SL is gaining importance in the National Building Code of Canada to determine the appropriate level quality for building materials, and as such, SLP should be included in building material standards [20,21], but a lack of information on degradation factors, material properties and methodologies makes it difficult to implement standards in this respect.

Nowadays, SLP must also account for climate change. Climate change is a slow process often considered a non-issue, but due to the generally long SL of buildings, it is most unlikely that the climate at the time of construction, and much later at demolition, will be the same. Therefore, it is critical to consider it in the development, design and production of any BEM [1,4]. For example, microclimates are very specific meteorological effects in absolute proximity of the product and include factors like relative humidity (RH), temperature, solar irradiance and air pollutants [22,23]. The changing climate can have a significant impact on the overall useful SL of materials [24]. As climate changes, weather effects become more unpredictable and could cause unforeseen consequences on BEMs as degradation rates could be accelerated. Simulations can provide estimates of possible future conditions and it is more important than ever to consider the possible ramifications of climate change on the building industry, and to manufacture, test and develop the use of products that can withstand these changes [2,25].

This paper provides a brief review of degradation mechanisms in polymeric materials, before a more focused overview of aging and durability of polymers in BEMs, with emphasis on the function and significance of degradation factors for cladding, air/vapour barriers, insulation, sealants, fenestration polymers and gaskets. This is followed by a

critical look at SLP methods for polymeric BEMs, which helps identify knowledge gaps and areas that require further research. It should be mentioned that in this paper, all the building code requirements are given from the National Building Code of Canada and projected climatic conditions are presented for Canada, but the literature studies are not limited to Canada and the authors tried to cover case studies from all over the world.

## 2. Degradation Mechanisms in Polymeric Materials

Polymeric materials can be classified into two main groups: plastics and elastomers. Plastics have low elasticity and low tolerance to deformation, whereas elastomers have a high degree of elasticity and great tolerance to deformation [25–28]. Plastics and elastomers react differently to aging. For example, in cyclic temperature variations, building materials will expand and contract causing an opening and closing of joints. Polymers with plastic behaviour do not follow the joints movement without failing, whereas elastomeric polymers do allow for such movement. For this reason, joint sealants are elastomeric, at least early in their SL [26,29]. The durability of a material in any structure is crucial, as most construction endeavours are expensive and frequent repairs or rehabilitation is not economical [30,31]. Degradation is significantly influenced by material composition/structure and environmental factors [8,25], and ultimately it is the material environmental resistance that determines durability [31,32].

Mechanical, thermal, chemical, electromagnetic and biological factors can greatly impact the SL of polymeric products and how they degrade [25,27,32–35]. The properties of polymeric BEMs change over their SL because of degradation that result from environmental loads (Table 2).

**Table 2.** Degradation factors that affect the SL of polymeric BEMs; Adapted from Refs. [25,27,32–35].

| Factor | Examples |
| --- | --- |
| Mechanical | Gravitation, imposed/restrained deformations, impact from hail, vibrations |
| Thermal | High and low temperatures, cyclic temperatures |
| Chemical | Water, solvents, oxidisers, acids, bases, salts |
| Electromagnetic | Solar radiation, electric current |
| Biological | Plant, fungi, microbial growth, animal-related erosion (rodents, insects) |

Not all aging mechanisms apply to every BEM. For instance, polyethylene, a common vapour barrier, is very sensitive to photo- and thermal-oxidation, but it is insensitive to hydrolytic aging. In contrast, nylon-6, also used as a vapour barrier, is more sensitive to these aging factors. Therefore, it is critical to have a solid understanding of material composition, possible aging mechanisms and in-service conditions to predict the performance and SL of polymeric BEMs.

In the examination of aged materials, the degradation mechanisms during both construction and service phases must be considered. However, the significance of degradation during construction is often disregarded because it is short compared to the service time, yet the environmental loads during construction can be much greater than the service loads. This includes vapour barriers (Figure 1), for example, which can be exposed to sunlight, heat and humidity peaks during construction but are only tested for water vapour permeance [36,37].

The modelling of degradation processes can help to predict the SL of BEMs [17]. Some considerations for modelling degradation mechanisms in BEMs include climatic load based on historical and expected future weather loads, building element functions and properties and relative movement of adjacent elements. These considerations can help to determine SLP [23].

### 2.1. UV Radiation

Solar radiation, particularly the UV portion of the solar spectrum between 200 and 400 nm, degrades organic materials by photochemical processes [38]. The extent of damage depends on the intensity of the UV radiation and the exposure time [24]. Most organic-based polymers undergo photolytic or photo-oxidative reactions due to exposure to UV radiation, and their useful life decreases [25,39]. Many polymers contain chromophores such as alkene (e.g., polybutadiene), ester (e.g., polyacrylates), or aromatic groups (e.g., polystyrene), which can absorb UV energy. When a chromophore absorbs energy, it is raised to an excited state.

If the polymer is unable to release the absorbed energy, then it may cause a free radical to be released, initiating photooxidation through a free radical reaction called autooxidation [39,40]. Initiation can commence in a number of other ways in polymers, but this is the primary source of UV degradation for polymers [39–42]. Free radical reaction occurs in three steps: initiation, propagation and termination. Initiation, as previously discussed is caused by the absorption of UV radiation raising the chromophores to an excited state, when the energy cannot be released, it results in scission. Propagation occurs when the free radicals mobilise with the aid of oxygen in the polymer matrix and abstract atoms from the polymer to attempt to stabilise, causing further destabilisation and producing more radicals overall. Termination is the last step of a free radical reaction and it occurs when two free radicals combine to stabilise, ending the propagation cycle [39–41]. These reactions can break bonds, cleave polymers into smaller fragments that can wash away, evaporate or cross-link with neighbouring segments and form new alkene groups that can further oxidise or cross-link, or both [25,42]. This reaction can be seen in Figure 2.

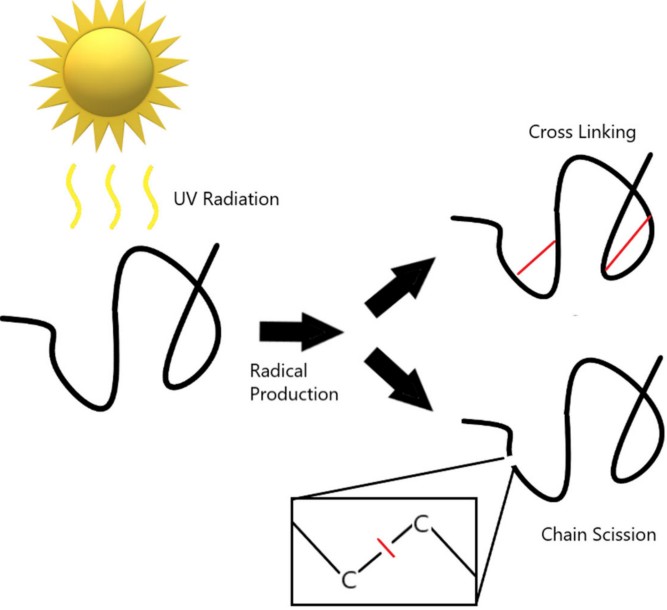

**Figure 2.** Polymer autooxidation by UV radiation.

These changes can be detected by physico-chemical methods such as solubility tests, thermal analysis, spectroscopy and gel permeation chromatography [25]. The initiation of these photo-chemical degradations eventually leads to material failure. The effects of UV radiation are exacerbated by elevated temperature, so as climate change causes an increase in temperature, the effects of UV radiation will increase [8,39].

UV degradation generally affects material's properties and performance. This includes discolouration, chalking and loss in surface gloss, which may lead to the aesthetic failure [43]. Failure may also come in the form of micro-cracking, embrittlement, increase in capacity to absorb water and loss of integrity by reduction in mechanical properties such as

tensile strength, impact strength and elongation [8,25,42,43]. The latter may be especially troublesome for sealants, which must remain flexible to follow the thermally induced opening and closing of façade joints.

Materials such as polyethylene (PE), polypropylene (PP) and polyvinyl-chloride (PVC) formulated for UV exposure contain UV light-stabilisers [40] to help prevent UV-related degradation. These additives also deteriorate over time, and eventually the polymeric material degrades. In opaque materials with inorganic fillers and pigments, degradation is often limited to the surface, because the inorganic material shields the bulk of the polymer [23,39]. From all of the polymeric components discussed in this paper, cladding, gaskets and fenestration are most commonly exposed to UV radiation. However, considering the construction time, each building envelope component can be potentially exposed to sunlight; therefore, UV degradation must be taken into consideration.

Artificial light sources are used to simulate UV exposure in labs [38]. The intensity of the radiation may be increased to enhance degradation and reduce the exposure time [44–46].

### 2.2. Moisture

Water damage is defined as the physical damage caused by moisture entering components through diffusion, exacerbating freeze-thaw effects, encouraging mould and mildew growth and reducing the effectiveness of BEMs. Degradation by moisture can also be caused by hydrolysis, or the reaction of a water molecule leading to the cleavage of a functional group [47]. This reaction occurs in polymers that have water-sensitive group in the polymer backbone, some polymers that are susceptible to hydrolysis are polyanhydrides and polyesters. The rate of degradation due to hydrolysis can vary from hours to years depending on the surrounding conditions, functional group, backbone structure and pH. Hydrolysis of semi crystalline polymers such as esters occurs is two stages, first water diffuses into the amorphous regions and then the moisture penetrates and degrades causing cleavage, this can be seen by the hydrolytic reaction in Figure 3.

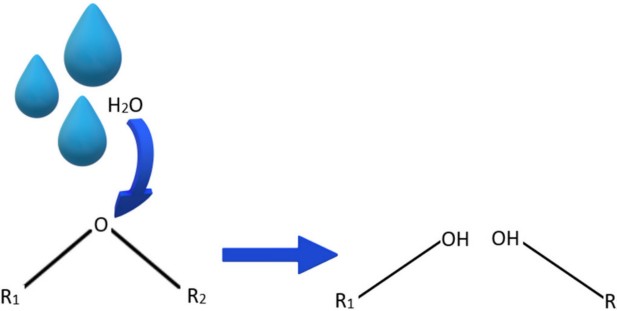

**Figure 3.** Ether hydrolysis reaction.

Moisture sources can be categorised into two primary categories, internal and external sources. Internal sources of moisture include showers, taps and cooking. External sources of moisture include rain, sleet, snow and runoffs [3]. In Nordic climates, internal sources are generally managed with vapour barriers near the interior side of the wall (Figure 1). External sources of moisture are managed with a water barrier, often with the combined use of exterior cladding and air barrier/water sheathing membrane (Figure 1). Because moisture exists on either side of the building envelope, all of its components can be potentially exposed to moisture. Water can deteriorate both vapour and air barriers over time. Damage to BEMs can arise from hydrolysis, where a water molecule reacts with water-sensitive chemical groups to affect the properties of the material, e.g., nylons, polyesters and polyurethanes (PUs). Hydrolysis is more significant in the presence of acid or base (sources from acid rain or alkalinity of the concrete, respectively), where they can act as catalyst to accelerate degradation [48].

The location of the component in the building envelope and its direct or indirect exposure to moisture will affect the risk of degradation [49]. For example, a polymeric

cladding exposed to hot and humid conditions will dry more quickly and be less affected by moisture-related degradation than a deeper layer that may become wet, such as sheathing or insulation.

Experimentally, water resistance may be assessed from sample immersion in water and calculations of weight gains, as described in ASTM D2842 [44,50,51]. The effect of rain can be simulated with water sprays as described in ASTM C1442 [52], whereas the high humidity of coastal conditions may be simulated with high humidity chambers [44].

### 2.3. Temperature

Thermal degradation occurs when thermal energy exceeds the dissociation energy of a chemical bond, the bond strength. Thermal energy leads to vibration in chemical bonds and when this vibration exceeds the bond strength because of excessive temperatures, the bond ruptures and material degradation follows [23,53]. Different bonds have different strengths [54], and consequently, thermal resistance depends on the material composition. However, thermal degradation does not occur in BEMs because service temperatures are well below thermal degradation temperatures (generally in excess of 200 to 400 °C). In practice, the thermal degradation of polymers occurs through thermo-oxidative degradation, the combined action of heat and oxygen. As such, the rate of oxidation will increase with temperature, and in simple cases only, it will follow the phenomenological Arrhenius law [54], based on which the rate of a chemical reaction doubles for each increase of 10 °C in temperature, as a rule of thumb [3]. Such a rate rarely occurs, however [49], partly because oxygen diffusion depends on material composition and thermal transition temperatures (melting and glass transition temperatures) [53]. Consequently, it is preferable to measure the actual oxidation or degradation rate, rather than using the Arrhenius rule of thumb. For instance, Lellinger et al. recently showed [55] that the rate of thermal aging of solid nylon increases about 10 to 200 times for a 70 °C increase in temperature, whereas the Arrhenius rule of thumb would have predicted an increase in rate of 128.

In summer, temperature extremes occur near the exterior of the envelope, so components located closest to the exterior are at a greater risk of thermal oxidation. Notwithstanding, temperature fluctuations and gradients exist throughout the envelope and thermo-oxidation can affect all components based on their composition and physico-chemical characteristics such as oxygen diffusion rates and thermal transitions.

In experimental considerations, a series of temperatures greater than service temperatures are used in the laboratory to expose materials to thermo-oxidation conditions. SL estimates are made from extrapolation to service temperatures based on the Arrhenius law or time-temperature shift factors [55], or non-linear methods [56]. Aging temperatures must not cross a thermal transition, which would make for inaccurate aging rate calculations [49].

### 2.4. Mechanical Stress

Some components of the building envelope may undergo continuous or momentary cyclic stresses. For example, wall cladding must be able to withstand changing wind pressures [57]. Common sources of mechanical stress include air pressure changes or intense winds/storms, expansion and contraction due to freeze thaw cycles [49], and expansion caused by intense heating [7]. Mechanical stresses such as tension, compression, expansion and bending will reduce the overall strength of a material [58]. Repeated stresses may lead to fatigue that will break chemical bonds [59] which may increase cross-link density, raise material stiffness and reduce flexibility. Another form of mechanical stress refers to loads on structural elements. However, most of polymeric BEMs are used in non-structural elements; one exception is glass fibre-reinforced polymeric (GFRP) brick ties, or masonry ties. Unlike the more common stainless steel ties, the GFRP ties are not susceptible to corrosion, but may be susceptible to other forms of degradation such as creep, the ties have lower ductility than the stainless steel alternatives and are more prone to complete fracture [60].

## 2.5. Biological Attack

Biological attack is the assault by a living organism, usually fungi, rodent or insects onto a material [61]. BEMs based on synthetic polymers are rarely the subject to biological attack, which is not to say that they never provide either a food source or shelter. Bio-deterioration generally requires moderate temperatures, an oxygen-containing medium, a water source and an adequate food supply [61], and therefore it is most common in landfills [62]. Rodents and insects can damage BEMs, and they are most common in the insulation layer [3]. Fungi can attack some polymeric materials, but common synthetic polymers such as thermoplastics and sealants are not normally sufficient sources of food. In contrast, polymer additives and biopolymers may be subject to biological attack, this includes for instance, wood as a glucose source, some phosphates used to increase polymer fire resistance and gypsum board paper in sufficiently humid areas. Although biological attack on polymers is not common, there is likely no material completely immune to biological attack [61]. Consequently, standard test methods do exist, which make use of specific fungi types, or insects like termites [63–65]. The bio-deterioration of architectural coatings, a material outside the scope of this review, has been extensively covered by Gaylarde et al. [66].

## 2.6. Synergetic Effects of Degradation Factors

In Sections 2.1–2.5, important sources of degradation for polymeric BEMs were described. Each can individually degrade building components, but they generally act together, sometimes synergically [67]. Consequently, aging tests with combination of degradation factors are required to create realistic accelerated aging conditions. For instance, thermo-oxidative aging is normally carried out in moisture-free conditions, but this is not representative of service conditions where moisture may accelerate other aging factors [44], including water driven hydrolysis [31]. Hence, for the thermo-oxidation studies of BEMs, the use of constant humidity conditions should be considered [31]. Such tests can be a challenge to design because the severity and frequency of each factor in relation to the others must be determined [44], and once this is done, several combinations of aging factor may be considered [68].

As a summary to Section 3, it may be stated that BEMs will experience several types and extents of degradation loads while in construction and service. Therefore, to determine how degradation mechanisms alter the properties of the BEMs, tests that combine degradation factors are used. In the next section, each polymeric BEM from Figure 1 will be reviewed for their main function, degradation mechanisms and related studies.

## 3. Polymeric BEMs, Function and Degradation

In the following section, the function of building envelope material is discussed, along with possible degradation considerations, and a summary of relevant studies, with a focus on typical approaches to assess material degradation and its relationship to performance. The scope includes electromagnetic (UV radiation), chemical (moisture), thermal, mechanical and biological degradations. These pathways may affect a single or all envelope components. A combination of degradation factors may lead to more rapid, severe or complex degradation mechanisms. For example, and as with thermo-oxidation mentioned in Section 3.6, UV radiation and oxygen act together to cause photo oxidation [34].

Not all degradation mechanisms in Section 3 apply to every BEM, because either a particular degradation mechanism is not significant, or it has not been studied separately from another. For example, UV radiation is considered a significant degradation mechanism for cladding but there are no studies on the effects of UV radiation alone on polymeric cladding because UV radiation is generally combined with other environmental loads. Table 3 shows the various loads and how they apply to each building component, either during the construction phase or throughout the SL.

**Table 3.** Environmental loads that apply to polymeric building envelope components.

| Load \ Building Component | Cladding | Vapour/Air Barrier | Insulation | Sealants | Fenestration | Gaskets |
|---|---|---|---|---|---|---|
| UV Radiation | Sunlight (installation and service) | Sunlight (installation) | Sunlight (installation) | Sunlight (installation and service) | Sunlight (installation and service) | Sunlight (installation and service) |
| Moisture | Rain, Snow, Ground Water, Dew. (installation and service) | Air Barrier: Rain, Snow, Ground Water, Dew Vapour Barrier: Showers, Tap, Cooking (installation and service) | Rain, Snow, Ground Water, Dew, Showers, Taps, Cooking (installation and service) | Rain, Snow, Ground Water, Dew (installation and service) | Rain, Snow, Ground Water, Dew (installation and service) | Rain, Snow, Ground Water, Dew (installation and service) |
| Thermal | Hot and Cold Weather, Building Heating and Cooling Systems (installation and service) | Hot and Cold Weather, Building Heating and Cooling Systems (installation and service) | Hot and Cold Weather, Building Heating and Cooling Systems (installation and service) | Hot and Cold Weather, Building Heating and Cooling Systems and service) | Hot and Cold Weather, Building Heating and Cooling Systems (installation and service) | Hot and Cold Weather, Building Heating and Cooling Systems (installation and service) |
| Mechanical | Wind, Air Pressure, Thermal Expansion/ Contraction (installation and service) | Air Barrier: Wind, Air Pressure, (installation and service) | External Insulation: Thermal Cycles and Gradients (installation and service) | Wind, Air Pressure, Thermal Expansion/ Contraction (installation and service) | Wind, Air Pressure, Thermal Expansion/ Contraction (installation and service) | Wind, Air Pressure, Thermal Expansion/ Contraction (installation and service) |
| Biological | Fungus, Rodent, Insects (unlikely) | N/A | Fungus (service) | Fungus, Rodent, Insects (installation and service) | Fungus (installation and service) | N/A |
| Combined | All | UV Radiation, Moisture, Thermal, Mechanical | All | All | All | UV Radiation, Moisture, Thermal, Mechanical |
| UV Radiation | Sunlight (installation and service) | Sunlight (installation) | Sunlight (installation) | Sunlight (installation and service) | Sunlight (installation and service) | Sunlight (installation and service) |
| Moisture | Rain, Snow, Ground Water, Dew. (installation and service) | Air Barrier: Rain, Snow, Ground Water, Dew Vapour Barrier: Showers, Tap, Cooking (installation and service) | Rain, Snow, Ground Water, Dew, Showers, Taps, Cooking (installation and service) | Rain, Snow, Ground Water, Dew (installation and service) | Rain, Snow, Ground Water, Dew (installation and service) | Rain, Snow, Ground Water, Dew (installation and service) |

### 3.1. Cladding Function and Degradation

3.1.1. Function

Cladding, also known as siding, is the façade of a structure, traditionally made of clay-fired brick or wood. The cladding material is used as the first line of defence against weather (rain, wind and snow). It also provides some level of insulation and protection for the inner layers of the building envelope [1]. According to the National Building Code of Canada (NBCC) [20], the required function of cladding is to minimise the ingress of precipitation into the assembly and prevent the ingress of precipitation into interior spaces.

Polymer-based siding has gained popularity over the last few decades due to ease of application, moisture resistance, low maintenance, low cost and energy efficiency related to

some insulating properties [1]. Siding of polyvinyl chloride (PVC), better known simply as vinyl siding, started to gain popularity in the 1950s, and today unplasticized PVC (PVC-U) is the only polymeric material mentioned in the NBCC as cladding that conforms to the requirements of CAN/CGSB-41.24 [69]. Table 4 compares the related thermal conductivity of some common cladding materials, and of the four products shown.

The only material with a lower thermal conductivity than PVC is wood. Polymer-based products also provide a series of modern looks and textures such as wood grains or a simple matt finish [70].

**Table 4.** Thermal conductivity of common cladding materials.

| Cladding Material | Thermal Conductivity (W/m K) | Reference |
| :---: | :---: | :---: |
| Wood (Traditional) | 0.04–0.12 | [71] |
| Brick (Traditional) | 4.81 | [72] |
| Aluminium | 205 | [71] |
| PVC-U | 0.13 | [73] |

Polymeric cladding have now gone beyond PVC-U to include glass fibre-reinforced plastics (GFRP), phenolics, wood plastic composites (WPCs) and filled polycarbonates, polyesters or PUs. Fibre-reinforced polymers (FRP) are yet another addition to the list. The most common FRPs use carbon, glass or aramid (nylon) fibres, in combination with polyester, vinyl ester or epoxy resins [74]. WPC cladding may be produced with PP, and PVC, with PP being the most common [75,76]. Material producers purport specific advantages to each product: for instance, GFRPs have high specific strength and a moderate cost [57]; solid or foamed PVC-U has the feel and workability of wood, good resistance to rot and warp; WPCs are stiff, impact, warp, thermal and rot resistant, with relatively low moisture absorption; and they are made with recycled polymers and wood scrap such as saw dust [77].

In Canada, these polymeric materials do not fall under standard CAN/CGSB-41.24, and beyond the claims, good material performance must be demonstrated and established. Polymeric materials, including cladding, have coefficient of thermal expansions that can be significantly larger than wood [78,79] and this must be accounted for in design and construction. Material producers must demonstrate that their innovative product meet the intent and durability requirements of the NBCC. This is most often done through the Canadian Construction Material Centre (CCMC), a government organisation that provides opinions based on material evaluations. Canadian building officials and architects often require CCMC evaluations to accept products in new constructions and renovations. The same is true of other innovative construction products of unproven performance.

### 3.1.2. Environmental Loads onto Cladding

The primary concerns for degradation in cladding, as stated in Table 3, include exposure to UV radiation, extreme heat and cold, elevated moisture or humidity and mechanical stresses. The cladding is the most external component in the building envelope, as shown in Figure 1, and as such, it receives the most exposure to sunlight, rain, wind, snow and temperature changes. The sun is a source of UV degradation and elevated temperature. Rain, snow and dew are all frequent sources of moisture for the building envelope, and when combined with elevated temperature, can cause hydrolysis. Moisture can also increase the impact of low temperatures in freezing. Wind and changes in air pressure will act on the cladding creating mechanical stresses. Biological degradation by fungus growth is less likely in areas with high amounts of UV radiation, and is likely to only occur in shaded areas with high moisture contents, therefore is not very significant in cladding. Chemical degradation can occur due to acid rain and the corrosion of mechanical attachments causing metal-mediated polymer degradation. Cladding is the building component that experiences the most exposure for the longest periods.

In Canada, cities can receive from 1567 (Sherbrook QC) to 2396 (Calgary AB) average hours of sunlight a year depending on location [80]. Riahinezhad, et al. [31] aimed to provide field data on actual aging conditions in Ottawa, Ontario so as to relate accelerated aging techniques and real world conditions, providing monthly average temperatures and RH data. The hottest months of the year were determined to be July and August averaging at 24.6 °C and 24.9 °C respectively, the coldest month of the year was determined to be January averaging at −2.5 °C. The absolute moisture content in the east-facing wall was calculated and for three years more than 6 g water/ $m^3$ air was present in the wall majorly, but often the moisture values were much higher. The components of the building envelope must be able to resist these conditions.

### 3.1.3. Degradation Studies
Mechanical

The case study performed by Mathieson and Fam [57] aimed to determine the performance limits of GFRP-faced sandwich insulation panels consisting of fibre-reinforced exteriors with lightweight PU foam cores used for cladding. Mathieson and Fam [57] conducted research on the effect of reversed bending fatigue on GFRP panels. The test was designed to apply four-point bending with the ability to completely reverse loading to test the resistance of these materials against strong wind pressure or suction. The panels then underwent fatigue cycles, these cycles were periodically halted to determine the change in stiffness. The cycles continue until fracture or failure of the panel or core occurred. It was determined that a force up to 23% of the control will never cause the panel to fail, meaning the panels can withstand any force under 3.32 kN. To put this into perspective, hurricane force winds can apply a pressure of 7.35 kN on a 10 $m^2$ wall, whereas average wind speeds of approximately 4 m/s will apply a force of 96 N on a 10 $m^2$ wall, meaning such panels can resist average wind pressures but should not be used in areas where extreme weather events such as hurricanes are likely [81]. No major change in panel stiffness occurred. Although this study provided valuable information on mechanical forces applied on claddings, it is not fully representative of outdoor aging conditions, as other degradation factors were not considered.

Combined Aging Factors

As mentioned earlier, a common cladding material is PVC-U. Isner and Summers [82] completed a study related to the effect of production conditions on the weathering rate of PVC siding, where the weathering rate was determined by the overall colour change as per ASTM D 1925-70. Three experiments were run to investigate the effects of different stabiliser amounts, production rate and melt temperatures on the stability of the product. The first experiment was run on white PVC-U siding with reduced stabilisers to determine the amount of degradation during processing. Samples were run at four different melt temperatures and production times, correlating with screw RPMs. Then the samples were weathered for 200 and 400 h in a carbon arc Fadeometer, the exact conditions of the Fadeometer were not indicated. The weathering resulted in an increase in the yellowness index of all the samples. The results for samples with the same production time (screw RPMs) suggested that an increase in melt temperature results in a greater change in colour; this can be overcome with a higher production speed. This was expected as stabilisers influence the thermal stability of the product. No outdoor exposure test was run with this type of siding to corroborate these results. The second experiment used blue PVC siding fluxed in a Brabender mixing head until rapid degradation occurred, deemed ultimate processing time. Specimens were prepared at different processing times and weathered outdoors for 6 months at Avon lake Ohio. This experiment demonstrated a direct correlation between equilibrium processing time and colour change. The longer the processing time the more the colour changes when weathered, while a shorter processing time results in better colour retention. The melt temperature did not ultimately have any effect on the colour changes in this experiment. The third experiment used tan colour PVC siding with

various stabiliser levels. The samples were weathered outdoor for six months at Avon Lake, Ohio. The results showed that the samples with less stabiliser underwent a more severe colour change. While colour change is an indicator of material aging (because physical appearance is a consideration in cladding), it is not indicative to overall usefulness.

As mentioned earlier, another common polymeric cladding material is WPCs. The study of the physical property of WPCs cladding is relatively new and it is gaining in popularity. It is purported that WPCs can retain the natural look of wood and resist degradation, but degradation does occur on exposure to moisture and sunlight [77,78]. The woody filler may swell on exposure to water and shrinking on drying, and cyclic exposure can lead to surface cracks and internal fracturing. Photo-oxidation is also said to degrade lignin in the filler and enhance further moisture-related damage [78].

WPC products have shown that very good marriage between polymer and filler is required for enhanced durability, which may be obtained with the addition of compatibilizers. For instance, Martikka, et al. [83] studied the effect of compatibilizers on UV resistance of high density polyethylene (HDPE), low density polyethylene (LDPE) and PP WPCs. Accelerated weathering tests were performed with a xenon-arc lamp with total exposure of 500 h (a cycle of 102 min of light exposure, followed by 18 min of light exposure and water spray). The accelerated weathering caused a lightening or bleaching effect in all the samples. SEM was used to determine any changes on the surface; the sample without any compatibilizer had more wood fibres near the surface covered only by a thin layer of polymer, compared to the samples with a compatibilizer that retained its surface integrity with less severe cracking. Fourier transform infrared (FTIR) spectroscopy analysis was used to examine the surface chemistry. The results indicated a clear formation of carboxylic groups, the only significant statistical change occurred in the sample without any compatibilizer and the sample with maleic grafted agents that lends to the possibility that the compatibilizer does not have a strong influence on UV degradation. Tensile strength was used as the determining factor for loss in mechanical performance; in general, there was no significant loss in tensile strength for any sample. The results suggest that the use of a compatibilizer does improve the overall durability; however, the impact is not great.

A research study performed by Friedrich [84] investigated the degradation of WPC claddings with PP, PVC and HDPE as the polymer matrix with pine or bamboo as a softwood component, or maple fibres as a hardwood component. The products were subjected to one year of natural weathering in Central Europe, where they were exposed to radiation, high and low temperature, precipitation and freeze-thaw cycles as indicated in Table 3. The durability of WPCs was studied by measuring the flexural strength and pull-through resistance of the screws placed in the composites. Radiation and freeze-thaw cycles had a significant negative impact on the flexural strength of the samples. However, fastener pull-through strength was not affected significantly by natural weathering. This suggested that the loss in flexural resistance alone cannot be a realistic indicator for durability and lifelong performance of WPCs.

Friedrich [76] also reviewed aging methods for WPCs used for cladding. The review split studies into artificial weathering and outdoor weathering methods. The studies that used artificial weathering were completed by Kallakas et al. [85], Beg and Pickering [86], Soccalingame et al. [87] and Stark et al. [88]. The studies that used outdoor weathering were carried out by Taib et al. [89], Homkhiew et al. [90], Hung et al. [91], Silva et al. [92].

Kallaks et al. [85] completed an accelerated aging study that focused on the impact of moisture absorption and UV radiation on the mechanical and physical properties of WPCs. Two thermoplastics and two types of wood flour were used for the WPCs: pelletised PP and linear low density polyethylene-grafted maleic anhydride (LLDPE-g-MAH), and birch and aspen ground into a flour. Three different coupling agents were used to improve adhesion, LLDPE-g-MAH, 3-amino-propyltriethoxysilane (APTES) and PVA. The samples had various polymer/wood ratios, such as 100% wood, 80/20 and 65/35. The UV radiation aging performed in accordance with EN ISO 4892-3:2006 [93]. The samples were placed in a UV chamber equipped with two UV lamps at ambient temperature and RH for a total

of 500 h. Another test was performed to analyse the water absorption, where the samples were immersed in distilled water for 3 weeks. Flexural strength, modulus of elasticity, impact strength and colour change were analysed for each sample. It was found that the mechanical properties depended mostly on the wood fraction size, and smaller wood fraction size gave better flexural properties. The coupling agent also influenced durability; the best results came from the wood flour and LLDPE-g-MAH samples. Adding wood flour to the polymers decreased the impact strength, increased water absorption and swelling, and the more water absorbed the lower the modulus of elasticity. UV radiation decreased the flexural modulus of elasticity and flexural strength, and resulted in a colour change in all specimens.

Another accelerated aging study was performed by Beg and Pickering [86] who studied the aging of PP and pine wood fibre composite. The samples consisted of pure PP, PP with bleached and unbleached wood fibre. The accelerated weathering had cycles of UV radiation for 1 h, water spray for 1 min followed by 2 h of condensation. The temperature was maintained at 50 °C. The samples were exposed to the aging for 150, 400, 600, 800 and 1000 h. Samples showed a change in colour and deposition of white chalky material after aging. The pure PP samples underwent a decrease in weight, and the composites had an increase in weight proportional to the weathering time. Failure strain, impact strength and Vicker's hardness number decreased for all samples after weathering. Overall, the bleached fibre composites were found to have higher tensile strength, failure strain and impact strength than the others.

Soccalingame et al. [87] studied the accelerated UV aging of PP mixed with PP-g-MAH and spruce wood flour. The aging consisted of a cycle with 102 min of UV exposure, at 60 °C and 65% RH, followed by 18 min of water spray, a total test time of 14 days. After aging, the samples underwent reprocessing to determine if mechanical properties could be recovered. The results showed an increase in elastic modulus, and a decrease in yield strength, impact strength and elongation, all of these properties were recovered after reprocessing. Infrared spectroscopy led to the conclusion that photo-degradation did occur on the samples and the decrease in mechanical properties is evident of aging. The reprocessing phase did lead to a recovery of many of these properties and demonstrated the potential usefulness for these types of products.

Stark et al. [88] aimed to determine the effects of accelerated aging on WPC. The first batch of samples were 50% pine wood flour and 50% HDPE by weight. In the second batch, two photo stabilisers were added to the WPC mixture, hydroxyphenyl benzotriazole (UVA) and zinc ferrite pigment. Some samples contained both stabilisers and others only one. Injection moulding and extrusion were used to create the samples. Accelerated aging consisted of two tests, the first was only UV radiation exposure for 2985 h, the second test consisted of 108 min of UV light, followed by 12 min of water spray and UV, repeated for a total of 3000 h. There was a lightening of colour for all samples, the samples exposed to just UV did not lighten as much as the samples exposed to UV and water spray. There were no major differences in colour change between the extruded and injection-moulded samples. The samples with both stabilisers showed the least change in colour. Injection moulded samples and samples with stabilisers exposed to UV only experienced an increase in the modulus of elasticity, and extruded samples experienced a decrease. The modulus of elasticity also decreased for all samples exposed to UV and water spray, extruded samples had more of a decrease than injection-moulded samples. All of the stabilised samples had a change in modulus of elasticity; no major difference was noted between stabilisers. Samples with both stabilisers had the lowest strength overall, the samples with the UVA stabiliser has the best strength. Ultimately, both photo stabilisers were effective in aiding mechanical property retention after weathering, the zinc pigment was more effective.

The first study focused on outdoor weathering reviewed by Friedrich [78] was performed by Taib et al. [89]. The samples consisted of HDPE and Meranti wood flour (WF) with Hindered amine light stabiliser (HALS) and UVA photo stabilisers. Five formulations were tested: 100% HDPE (1); 50% HDPE and 50% WF (2); 49.5% HDPE, 50% WF and

0.5 % UVA (3); 49.5% HDPE, 50% WF and 0.5% HALS (4); and 49% HDPE, 50% WF, 0.5% UVA and 0.5% HALS by weight (5). Outdoor exposure occurred for a total of 2000 h at Universiti Sains Malaysia. Samples were analysed at 500, 1000, 1500 and 2000 h. Outdoor climatic conditions were tracked regularly. The samples were analyzed by SEM, FTIR, colour measurements and flexural properties. The addition of UVA and HALS to the samples did prevent surface cracks, as less damage was evident on those samples than the sample without stabilisers after weathering. FTIR showed evidence of chain scission for all samples. Weathering also resulted in lightning of all samples except for the pure HDPE sample which darkened with weathering. Sample 1 had the worst discolouration and sample 5 (HDPE/WF/UVA/HALS) had the least discolouration. Flexural strength ultimately decreased for all samples after 2000 h of weathering. Overall it was determined that UVA worked best as a stabiliser for samples of this type.

Homkhiew et al. [90] studied the effect of outdoor weathering on recycled and virgin PP and rubberwood flour (WF) with various polymer to filler ratios. HALS stabiliser was also used in some samples. The composites were weathered outdoors in Thailand for 360 days in wood exposure racks according to ASTM D1435-03 [94]. The colour, hardness and flexural strength of the samples were analysed before and during weathering at 60, 120, 180, 240, 300, 360 days. There was an overall increase in colour lightness; the colour of the samples without fillers increased more, and samples with fillers fluctuated between lightening and darkening in colour over time. The hardness of the samples decreased, the unfilled virgin samples only began to significantly decrease after 120 days, the unfilled recycled samples decreased more than the virgin samples. The samples with fillers had smaller decrease in hardness. SEM showed that the composites had smooth surfaces before aging, after aging large surface cracks appeared. Samples that were 25% WF exhibit less cracking than those that were composed of 45% WF. The recycled samples had more severe cracking than the virgin ones. There was deeper cracking present on composites without UV stabiliser. The flexural strength of unfilled samples decreased greatly over time, samples without stabilisers had a more severe decrease. Samples with high quantities of WF had a greater decrease in flexural strength. Overall samples with less WF did not resist weathering as well as samples with more WF and the use of UV stabilisers aided in weathering resistance.

Hung et al. [91] examined the outdoor weathering of HDPE and Makino bamboo composites. Some bamboo was treated with acetic anhydride and dimethylformamide (DMF), with 17% grain treated (WPG 17), 8% (WPG 8) and without (WPG). The composites consisted of 60% bamboo and 40% HDPE by weight. Samples were aged facing south and inclined at 45° at National Chung Hsing University for 1080 days, where temperature and RH were monitored regularly. At the end of the outdoor weathering period, the WPG 17 experienced the greatest colour change where the WPG experienced the least change. The mechanical properties of all the samples varied as a function of time. The flexural strength and modulus of elasticity retention ratios decreased significantly with weathering, the modified samples exhibited better retention ratios. Overall, the treatment of bamboo is a useful method to improve the weathering properties of composites, the sample with 17% treated bamboo flour showed the best retention of modulus and flexural strength.

The final study reviewed by Friedrich [76] was performed by Silva et al. [92] which examined the weathering of composites made up of wood flour from Mezilaurus itauba (ITA) and PP with ethylene vinyl acetate (PP/EVA). Samples consisting of PP/EVA and ITA, as well as pure PP/EVA were tested. Weathering was performed in accordance with ASTM D1435 [94], the samples were aged facing north at a 45° angle at Porto Alegre city in Brazil for 4, 8, or 12 months, where average UV radiation index, temperature and rainfall were monitored. The mechanical properties of the composites were analysed in accordance to ASTM D638 [95]. Overall, the PP/EVA/ITA samples maintained a higher fraction of their original mechanical properties after natural aging indicating that the addition of ITA causes them to be more stable and improves their resistance to weathering. SEM also

showed cracking for all samples after aging, the PP/EVA/ITA samples had more cracks but were not as deep as the PP/EVA samples.

The review by Friedrich [76] focused on the accelerated aging methods and their relevance to estimate the long-term degradation of WPCs used in cladding. This review highlighted the fact that there is less research focused on cladding and research on WPCs weathering focuses more on decking or fencing materials. He concluded that accelerated laboratory aging and the long-term natural weathering described above led to similar aging of composites. Ensuring similarity in aging is a clear and necessary step in the validation of accelerated aging methods and SLP. The only difference between the methods is related to the intensity of aging factors, which is due to the unpredictability, type and length of exposure time that materials receive during long-term natural weathering. The review by Friedrich [76] concluded that accelerated laboratory aging can be a reliable technique for determining the effect of environmental factors on the stability of the materials.

### 3.2. The Air and Vapour Barriers: Function and Degradation

3.2.1. Function

An air barrier system (ABS) or assembly is a collection of air barrier materials and accessories, and can be made up of many components including sheathing membranes (also called weather resistive membranes), sealants and tapes to create a continuous barrier for the building envelope [20,96,97]. In effect, an ABS is to prevent air flow across all building envelope cavities, including panel joints, floor to wall joints, openings in the wall for windows, ducts, pipes and the like. Often times, there is confusion between air barriers and vapour barriers. A vapour barrier aims to prevent the movement of moisture, and the air barrier, the movement of air [97,98]. This confusion is exacerbated by the fact that a single material can be defined as both air and vapour barrier, leading to the assumption that they are the same component with the same function [96,97]. In this paper, the 'air barrier' refers to the polymeric sheathing membrane used as part of the ABS (Figure 1), other relevant components such as the sealants are described in other sections.

An air barrier is used primarily to prevent external air from entering the interior of the structure. It prevents drafts and convection through the walls, averts mould and mildew and improves air quality and energy efficiency [2]. An effective air barrier has low permeability, high strength to withstand air pressure loads and is durable [99]. As the main purpose of an air barrier is to be air tight, the importance of an adequate ventilation system in structures with a good air barrier needs to be highlighted. The air quality of the interior depends on flushing out the internal pollutants. Sick building syndrome is of particular concern when adequate ventilation is not implemented [100]. According to the NBCC [20], the air barrier materials as part of the ABS must have an air leakage less than 0.02 L/(s·m$^2$) at an air pressure difference of 75 Pa, when tested in accordance to ASTM E 2178 [101], or conform to CAN/ULC-S741 [102]. With respect to ABS, the NBCC allows an air leakage ten times greater, 0.2 L/(s·m$^2$), to account for leakage at wall openings such as windows, electrical outlets, pipes and mechanical attachments. Typical air barrier materials are PP, PU (closed celled) and PE.

Sheathing membranes are the second line of protection behind cladding, the protection is more specifically against wind-driven rain and the retention of water that may collect behind cladding. The sheathing membrane must prevent incidental rain penetration at brick tie locations, other cladding attachments and at panel joints. The NBCC [20] requires that the outermost layer of a wall assembly, the sheathing membrane, be 'breathable' to allow moisture to dissipate during the drying season. Therefore, the NBCC has restrictions on the outermost layer having both low water-vapour permeance and low air permeance materials installed on the cold side of the assembly.

In contrast to the air barrier, the role of the vapour barrier is to prevent the transfer of moisture through the building envelope, generally from the inside of the dwelling into the wall cavity [5,6,96,103]. To effectively function, the vapour barrier is placed on the warm side of the insulation to limit condensation [7,96]. The vapour barrier does

not need to be completely continuous as unsealed laps, pin holes and minor cuts do not significantly affect the overall diffusion of vapour into the wall cavity. However, it is best to avoid these imperfections when possible as they may propagate other forms of damage or degradation [96]. In general, the water vapour migration through walls arises from two mechanisms, convective moisture transfer that includes the process of diffusion of water vapour and air through wall components and from advective transfer of moisture as the results of air movement through openings in the wall assembly [104]. The direction of flow and the rate of water vapour diffusion is a function of the vapour permeance of the material and the magnitude of the difference in water vapour pressure across the barrier [104].

Moisture is known to cause damage to the building envelope, as a result, there are three basic steps used for moisture control: control of moisture entry, control of moisture accumulation and removal of moisture [105]. Different climates will require altered strategies for moisture control with respect to the placement of membranes. For cold climates, assemblies require protection from interior moisture, so air and vapour barriers are installed towards the interior of the building envelope. In hot climates, the building envelope needs to be protected from the exterior moisture, this requires the vapour and air barrier to be placed closer to the exterior [105]. Water management is one of the most important factors related to building longevity as 75% of building failures are caused by water, and structures with an increased moisture content have a 2–9% increase in heat loss [98,106,107]. Moisture is produced as a natural by-product of internal building use as well as many external sources. Different spaces will have different levels of humidity and will have different vapour barrier needs. For example, an indoor swimming pool will have a much higher RH than that of a storage warehouse and therefore will require a better vapour barrier. No material is completely impermeable to water, so the goal of the water vapour barrier is not to eliminate vapour diffusion all together, but rather to control it [20,106,108]. According to the NBCC, vapour barriers placed on the interior side of the wall cavity (Figure 1) should not have a permeance greater than 60 ng/(Pa·s·m$^2$) to ensure proper control of diffusion [20]. Lower permeance may prevent drying of the wall cavity and lead to damage [103].

Without the vapour barrier or with ineffective vapour barriers, the moisture will travel through the wall into the insulation and the structural components. Water has a high thermal conductivity and the presence of moisture in the insulation will cause the insulation to be less effective, especially if external temperatures are below freezing, this could cause the moisture to freeze making the insulation less effective and creating a large drop in energy efficiency [5,106]. Transfer of moisture into the insulation can also create mould. Mould spores may be harmful to health and could cause otherwise preventable illnesses [109]. If moisture were to get into the structural components of the wall, serious structural damage could also occur such as wood decay and/or corrosion of metal fasteners used in the wall structure. The decay or corrosion of structural components relates to durability issues that may affect the serviceability of the structure and in time compromise safety. In addition to the mould-related issues, in areas where the temperatures drop below freezing, the moisture will freeze and expand, this expansion will put extra strain on the components which will lead to cracks or other forms of structural damage [6,110].

With a proper vapour barrier, similar to air barrier, an adequate ventilation system is important. With limited paths for moisture to escape from the interior of the house, the humidity level in the building may be high, and volatile organic carbons (VOCs) from paint, carpets and other furnishings may be trapped in the interior air. Hence, vapour-tight envelopes may decrease the overall air quality and increase the risk of the sick building syndrome [100,111,112]. Buildings with airtight envelopes need to be well ventilated to reduce the concentration of VOCs [111]. Typical materials used for the vapour barrier are HDPE and nylon-6.

### 3.2.2. Environmental Loads onto Air and Vapour Barriers

Table 3 describes the primary environmental loads for air barriers. They experience a loss in resilience and strength, and rupture due to oxidation, fatigue from air pressure and moisture contact. Degradation of sheathing membranes is caused primarily by prolonged exposure to elevated temperature, particularly in the presence of moisture [23]. Primary factors for degradation during installation include UV radiation, elevated temperatures and mechanical aging. Mechanical stresses will occur with the wind, the contraction and expansion of surrounding building components caused by changing temperatures. Biological degradation is not significant for the air barrier. Alkaline environments are not common in the building envelope except in the case of a material being applied on or next to a concrete surface.

The vapour barrier is susceptible to many of the same factors as the air barrier; the primary concerns during service are still elevated temperature and moisture. Elevated temperature will not be as significant near the interior as the temperatures are less extreme, being regulated by internal heating systems; a typical range of temperatures for the interior of a building is 19 to 22 °C, and the indoor humidity level should be 30 to 50% [113–115]. Moisture is significant as it can accumulate due to condensation, running water in showers and taps and vapour created by cooking. Chemical degradation caused by alkaline environments is not as significant of a factor, although contact with fresh concrete has been known to accelerate aging in some vapour barriers including PE due to the contact with moisture in concrete (known as alkaline hydrolysis) [115–117]. Biological degradation is not significant for the vapour barrier. During construction, the vapor barrier could be exposed to the outdoor and as a result, UV radiation and elevated temperature are significant factors in degradation.

The air and vapour barriers are discussed simultaneously in this paper due to the flexibility in their use. Often the location of these components changes and the same component can be used as both the air and vapour barrier, as such many of the studies discussed in this section were completed under the assumption that the component in question was both the air barrier and water-sheathing membrane. This is a common occurrence for Tyvek water-sheathing membrane or similar products.

### 3.2.3. Degradation Studies
#### UV Radiation

Marston completed a study [8] with LDPE and PP, where these materials were used primarily in air and vapour barriers, two types of LDPE, a transparent film and one infused with carbon black and one clear PP. The clear PP sheets showed the most discolouration and decline in mechanical properties. After three years, the materials had extensive micro cracks and had lost all the mechanical strength. The clear LDPE lost over 65% of its mechanical strength due to micro cracking. The black LDPE exhibited no change in colour, and although it had a high density of fine micro cracks on the surface, it retained its mechanical properties [8]. These results demonstrate clearly the mechanisms of UV radiation on polymeric materials, the brittleness and loss of strength cause by the cleavage of bonds and increased crosslinking and how opaque polymers are more resistant to this form of degradation.

#### Mechanical

Young and Meyer [7] studied fluid-applied air and water barriers and their elastic properties from freeze-thaw cycles and temperature variations. Membranes including silyl-terminated polyethers, silyl-terminated PUs, acrylics, butyl rubber and silicone rubber were tested for elongation at break and percent recovery at room and elevated temperatures. The results suggested that membranes behaviour is strongly dictated by their composition. For example, the silyl-terminated polyether membrane demonstrated the best recovery after 300% elongation at room temperature and the acrylic membranes had the most dramatic size change overall after the freeze-thaw cycles. In general, upon heating, the majority of

the membranes experienced a loss in elongation, and high ultimate elongation did not necessarily mean improved elastomeric recovery. Yet, a key aspect in the performance of fluid applied membranes is their elastic recovery and ability to return to their original shape when stress is removed.

Combined Aging Factors

Möller et al. [110] examined the 15-year degradation of a LDPE barrier installed in a test hut in Sweden. The barrier acted both as an air and vapour barrier and contained an antioxidant to improve durability. Two different field-aging environments were studied: (a) inside the wall, and (b) outside the wall, with a portion of the barrier hanging over the plasterboard exposed to the interior building air. After 15 years, the film inside the wall had a considerable loss of antioxidant compared to the film outside the wall, exposed to the interior environment. The loss of antioxidant was attributed to its migration towards the surrounding construction elements. Despite the lower antioxidant content of the barrier aged inside the wall, its oxidation induction temperature—the temperature at which a material starts to oxidise—was higher than for the barrier aged indoor. This was contrary to the expectations. Tensile test for barriers aged inside and outside the wall showed similar values, which indicated that there was very little LDPE degradation from this type of aging. The measurement of polymer chain size from size exclusion chromatography confirmed the absence of degradation during the 15 years of aging.

In another study, H. Orr et al. [112], reported on the performance of PE air/vapour barriers in six highly insulated houses in Saskatoon Canada. Inspections were performed to look for signs of damage to the barrier, a blower-door test was conducted to determine air tightness, and an energy assessment was conducted to identify areas of heat loss. The tests determined the airtightness of each house ranging from 0.78 to 2.55 air changes per hour (ACH), two of the six houses had a reference value from when the houses were constructed or renovated, and it was found that no significant decrease in air tightness could be detected. All houses had excellent heat-retention scores, and the main sources of heat loss were always windows, ventilation systems and basements [112]. Very little degradation of the air/vapour barriers could be detected after several decades of use. No material property testing was completed, because these barriers were in active use and could not be extracted. Both the studies of Möller et al. [110] and Orr et al. [112] demonstrated the good durability of PE barriers in active use over at least 15 years.

In a FTIR study, Jell and Nilsen [24] compared the oxidative effect of different accelerated aging methods on PP and HDPE membranes traditionally used as vapour barriers in Sweden. The methods were (a) QUV: 50 °C exposure with UVA and UVB radiation intensities of 28 W/m$^2$ and 2.8 W/m$^2$, respectively; (b) heat aging at 90 °C; (c) Nordtest Method NT Build 495 [118] which consists of a vertical climate simulator with four climate zones. From top to bottom: (1) IR/UV, with UVA and UVB intensities at 15 W/m$^2$ and 1.5 W/m$^2$, respectively; (2) water spray; (3) freezing; and (4) ambient. The samples spent 1 h in each zone in the order from 1 to 4. For short, this last method is called Simul. The extent of oxidative degradation from the three methods was reported on the basis of carbon oxidation, measured as carbonyl (C=O) absorbance peaks in infrared spectroscopy spectra [119]. The peaks for PP were larger than the peaks for HDPE for all the aging methods. This indicated that PP degrades more readily than HDPE, which is consistent with the ease at which free radicals form in these polyolefins [120]. Amongst the three aging methods, QUV led to the greatest rise in carbonyl peaks, due to the greater UV exposure from this method. In the other methods, the UV exposure was periodic. As a result, heat aging caused the least oxidation, and the Simul test was intermediate, which also led to surface cracks from freeze-thaw cycles. Hence, even though QUV resulted in the greater degradation, based on carbonyl intensity alone, the Simul test was more representative of actual field aging conditions in Sweden.

Jakubowicz and Klaesson [117] determined the effect of wet concrete and its high alkalinity on PE films. Two types of concrete were used, a standard K25 and one with

either 0.5% and 1% iron sulphate to see the effect of iron salts on PE degradation. PE films were aged at 70, 80 or 90 °C for up to 100 days in contact with solid concrete and in water solutions of deionised water (control), saturated CaO, 5% NaOH, 0.5% FeSO$_4$ and a mixture of salt similar to that in concrete. The solid concrete was either young or old; young concrete is more alkaline. Aging was measured based on the time to a 50% decrease in the PE elongation at break. The study showed that the high alkalinity of young concrete was more aggressive to PE than old concrete. In contrast, during solution aging, iron sulphate provided some protective effect. In being exposed to various aging media at 90 °C, PE film aged increasingly rapidly in this order: moist air, aqueous (aq.) FeSO$_4$, aq. salt mixture, aq. NaOH, aq. CaO, deionised water, fresh wet concrete. The authors concluded that metallic ions in concrete do have the potential to accelerate the aging of PE in construction applications where PE films is in contact with concrete, such as vapour barriers, membranes, floor underlay and ground sill insulation.

### 3.3. Insulation Function and Degradation

### 3.3.1. Function

Insulation is located between the vapour and air barriers in the building envelope. The primary function of insulation is to prevent heat transfer through the wall, and therefore it has low thermal conductivity [109,121]. Most polymers have lower thermal conductivities than metals, wood, concrete or other traditional building materials, which makes polymers excellent for use in insulation [1]. Proper insulation is key to improve the energy efficiency of buildings; it allows for more efficient and cheaper heating and cooling [1,122]. There are many benefits to the use of thermal insulation, for instance, a lower reliance on mechanical and electrical systems, greater energy efficiency, thermal comfort, structural integrity, fire protection, reduced energy costs, noise and environmental footprint [109].

There are many different insulation forms, which include blanket insulation, loose-fill, spray-in, pour-in, rigid boards and vacuum insulation panels (VIPs) [1]. In most cases, insulation is a two-phase system that consists of a dispersed gas in a matrix. The gas may be air or a gas with low thermal conductivity, and the matrix may be a fibre or a polymer. Fibrous materials are most often cellulose, glass or rock [123], whereas polymers are commonly expanded or extruded polystyrene (EPS and XPS respectively), and PU or polyisocyanurate foam [122]. Rigid insulation panels may be covered with a heat-reflective layer, and this is the case with VIPs, which have the lowest thermal conductivity of any insulation technology today, and for this reason, it is often considered the insulation of the future [121]. There are three primary forms of vacuum-based insulating systems: VIPs, vacuum-insulating sandwiches (VIS) and vacuum-insulating glazing (VIG) [124]. VIPs are the most common to the building envelope and consist of a porous material vacuum sealed in an envelope resistant to gas diffusion [115,118]. VIS have the same basic structure as VIPs, the most significant difference is that VIS have a stainless steel envelope, the envelope has a lower permeability, can withstand higher stresses than VIPs and is used primarily in systems under high pressure, such as submarines or in aerospace industries [125]. VIG is a method of glazing windows to improve the overall thermal resistance of the glass [126]. VIS and VIG are beyond the scope of this review.

### 3.3.2. Environmental Loads onto Insulation

Insulations may be vulnerable to heat aging, moisture uptake and biodegradation which may cause a decline in thermal performance, strength and rigidity over time [23]. Insulation is used to prevent the transfer of heat through the building envelope and as such will experience elevated heat on its outermost side. In cases where the vapour barrier is not effective, moisture can condense and accumulate in the insulation. Insulation may also become wet when the exterior water-sheathing membrane fails. Wet insulation has low thermal performance [127], and moreover the presence of water might damage material components susceptible to the hydrolytic action of water. Likewise, chemical aging may be catalysed by the corrosion of mechanical attachments [128]. Biological degradation must

also be considered, and it might occur in biopolymers. During installation, UV radiation, elevated temperatures and moisture are all significant degradation factors. In some scenarios, the insulation will be located closer to the exterior of the building envelope and therefore UV degradation and mechanical degradation caused by freeze-thaw cycles need to be considered.

### 3.3.3. Degradation Studies

Moisture

In this section the term degradation does not indicate a chemical change in the insulation, but rather a degradation in material performance, that is, an increase in thermal conductivity with moisture uptake.

Hansen et al. completed a study on the effect of RH on the thermal conductivity of cellulose insulation [129]. Four products were tested after storage in a chamber of 0 to 100% RH. In all cases, thermal conductivity remained fairly steady, with a rise of 5–10% in the absence of condensation, until RH reached 90%, where condensation led to a 50% increase in thermal conductivity.

In a review of 56 laboratory and field studies, Cai et al. [51] examined the effect of moisture on EPS and XPS thermal performance in an attempt to establish consistencies. Moisture uptake methods were isothermic, immersion in water, soil, or constant RH; and gradient methods, namely, the cold plate and cold-hot box, where a temperature gradient within the sample drives the diffusion of moisture. Aging in thermal insulation is often equated to a loss of thermal conductivity, and in the case of EPS and XPS, Cai et al. [51] showed that below 10% moisture by volume, thermal conductivity is not greatly affected. Above this level, however, conductivity increases rapidly with water content to a level that becomes unacceptable. The authors pointed out that EPS generally absorbs more water than XPS because of the skin on the XPS surface, and that isotherm tests lead to faster moisture absorption than gradient methods. This should not be surprising given that samples are respectively in contact with water in the liquid phase and the gas phase. Likewise, laboratory methods lead to faster water uptake than field methods, which was attributed to the constant nature of the laboratory test, in opposition with the cyclic nature of field conditions. It was further pointed out that laboratory test results do not correlated with field test results. This is arguably the most important reason for the lack of SLP with respect to XPS and EPS insulation in building envelopes.

Thermal

In this section, the thermal degradation and aging of insulation is discussed. As such, it does not only refer to chemical degradation, but also to the aging and deterioration of material properties, such as thermal resistance.

Closed-cell insulation foams are produced with a blowing agent, a gas of low thermal conductivity [29]. Over time, the outwards diffusion of this gas, balances with the inwards diffusion of air and increases the thermal conductivity [130]. Aging rate increases with a rise in temperature because diffusion rates increase with temperature. Cell walls thickness and density also affect the diffusion rate. Page and Glicksman [130] determine the diffusion rate of several blowing agents and determined that carbon dioxide had the largest diffusion coefficients whereas 2,2-dichloro-1,1,1-trifluoroethane had the lowest amongst those tested.

Mukhopadhyaya et al. [131] examined foam aging brought about by gas diffusion on the thermal performance of three polyisocyanurate foam insulations. The insulations boards had facers expected to reduce gas diffusion. Laboratory tests were performed on full thickness samples and samples cut thin from which the expected long-term thermal resistance could be calculated based on ASTM C1303 [132]. The initial heat transmission of the full thickness material was determined in accordance with ASTM C518 [133], and then after one year of conditioning at normal conditions of 24 °C and RH of 50%. The thin samples were exposed to normal conditions and were tested periodically for thermal resistance. The results were compared to those for the same products aged for 6 years

in a test hut located in Ottawa, Canada, and where normal conditions existed within the hut. Field samples were also placed on east- and west-facing wall to compare the effect of orientation. From the work, the authors concluded that thermal performance within a polyisocyanurate boards can vary significantly, facers do not prevent gas diffusion and the associated decrease in thermal performance, purported performance does not always reflect long-term field performance, and standard test methods may be improved to better compare with field test results. The work of Mukhopadhyaya et al. [131] thus demonstrates that standard methods are often perfectible and that laboratory work needs to be validated with field work to provide for better accuracy in the prediction of performance.

In a seminal paper, Norton [134] examined the outward diffusion of a low thermal conductivity gas and the change in gas composition within a foam. For this purpose, the foam was aged for 55 days, either in laboratory air or in sealed ampule. The focus was on $CFCl_3$, a chlorofluoro carbon and a PU foam. Gas composition was measured by mass spectrometry. Based on chemical analyses, gas diffusion rates were established and foam thermal conductivities could be predicted. The work of Norton [134] was an original effort to predict the rate of gas escape from foams and also the aging mechanism in foams blown with a low thermal conductivity gas. The work focused on foams for refrigeration units, but it applies equally well to thermal insulation for construction applications. Under typical field conditions, the blowing agent diffuses out of the foam to be replaced with gases from air. However, the replacement gas is not typical of air composition, with the oxygen/nitrogen ratio being 1:2 rather than 1:4, which shows that polar gases (oxygen) can better diffuse through some foams than less polar gases (nitrogen). This needs to be considered in the study of material oxidation and associated chemical degradation rates.

Jiao and Sun [135] studied XPS thermo-degradation in high temperatures by thermo-gravimetric analysis to calculate the activation energy in a non-oxidising atmosphere of nitrogen and an oxidising atmosphere of air. The activation energy for thermo-degradation in nitrogen was almost twice as large as that in air, which demonstrated that XPS degrades more readily in the presence of an oxidant [135]. Two factors must be considered with respect to high temperature oxidation studies of BEMs. First, the degradation and test temperature should not cross the thermal transition temperatures, which negates the use of the measured activation energies and the associated Arrhenius approach discussed earlier in Section 3.3 to determine SL in service temperatures [54]. Second, non-oxidising atmosphere are not representative of service conditions, but oxidizing atmosphere other than air may be useful, for instance, air pollutants like ozone, nitrous oxides and also saline air representative of coastal areas.

Biological Degradation

Rose and McCaa [123] examined the moisture uptake of fiberglass and cellulose insulations with and without vapour barriers. Cellulose is a biopolymer and it has the unique condition of being more vulnerable to mould growth than other synthetic polymers. Three configurations with cellulose were tested, one with a PE vapour barrier and some openings, one with PE vapour barrier and no openings and one with no vapour barrier. The configurations were set for an entire winter and the moisture content was measured in three different ways every hour. The moisture content of the first two configurations varied only slightly, whereas the moisture content of the configuration with no vapour barrier was up to 4% higher than those with vapour barriers. There was no severe damage to the insulation in the configurations with vapour barriers, however, the configuration with no vapour barrier exhibited severe corrosion on metal fasteners, caking of cellulose against the sheathing, and most noticeably, uniform mould growth from top to bottom of the insulation. The work of Rose and McCaa [123] highlights the importance of a proper vapour barrier especially in the case of materials sensitive to moisture and biological degradation. No information regarding the effectiveness of the insulation before or after degradation was provided.

Godish and Godish [136] completed investigations for mould growth in four building that had been insulated with wet spray-applied cellulose (WSACI). Three of the buildings were single-family residences (A, B and D) and one was an agricultural implement repair shop with attached living quarters (C). The exact geographic location of the buildings was not given in the study, though academic affiliations of the authors suggest that they were in the USA. Information on building age or weather conditions was also missing from the study. WSACI samples were taken from multiple locations including the exterior walls, ceiling overspray and interior walls of each house. The sample were tested for a variety of mould species. Some level of mould growth was found in all samples, and anywhere from 8 to 24 species were identified in notable concentrations. The highest concentrations of mould were observed in samples from building A and building B. Building A had notably high concentrations of the toxigenic Stachybotrys chartarum. This study confirmed that fungal contamination of WSACI had occurred for the four buildings in this study.

Combined Aging Factors

Stephenson et al. [137] evaluated the long-term performance of polymeric closed cell PU foam and EPS insulation materials exposed to humid thermal aging in the laboratory. The EPS was tested with two different blowing agents, the PU was not. The two products were aged for 5 weeks at 66 °C and RH from 30 to 90%. The physical appearance and the heat resistance of the samples were studied on a weekly basis. The PU showed no major physical deformation, but a colour change and slight blistering; thermal resistance also decreased by 27%. The EPS demonstrated no major physical changes, but after aging at 90% RH, it showed some mass loss. The thermal resistance of the EPS varied depending on the blowing agent. Decrease in the samples conductivity and mass was presumed to have been caused by the loss of the blowing agents and the infusion of other gases. After accelerated heat aging, the samples were left at ambient conditions for several months. The PU aged at the highest humidity recovered some thermal resistance, whereas that aged in low RH samples continued to decrease. The loss of thermal performance was attributed to the gas exchange throughout the foam. The work of Stephenson et al. [137] highlights the combined action of heat and moisture on the thermal performance of insulation. PU is clearly more affected than polystyrene, which likely reflects the hydrophilic nature of the polar urethane linkages compared to the hydrophobic and non-polar nature of styrene units.

Yuan [138] studied the change in thermal resistance of open- and closed-cell spray PU foams after various aging methods to accelerate gas diffusion in and out of the foam. Air can diffuse relatively quickly through open cell foam, in contrast to closed-cell foam, where cells encase a blowing agent. As a result, open cell foams have a higher thermal conductivity than a corresponding closed cell foam. Three aging methods were used on the closed cell foam (a) 23 °C and 50% RH for 180 days, as per ASTM C1029 [139] (b) 60 °C in dry air for 90 day, as per ICC-ES AC 377 [140] and (c) thin slicing as per standard CAN/ULC-S770 and storage at 23 °C and 50% RH, which is intended to simulate 5 years of aging in closed cell foams [131,141]. Method (b) alone was applied to the open cell foam. After aging, the thermal conductivity of the open cell foam had changed very little, 1–3%, which confirmed the thermal stability of these foams to diffusion-related aging. The performance of closed cell foam also changed little in aging after methods (a) and (b), but thermal resistance decreased by 13% in method (c) due to the fast outgoing diffusion of blowing agent in thin sections. Considering the work of Mukhopadhyaya [131], it may be said that heat aging is much less effective than thin slicing to estimate the 5 year thermal performance of closed cell PU foams.

Yrieix et al. [142] investigated the aging of VIPs. In so doing, the authors listed in exemplary manner eight possible applications for VIPs in construction, and for each of these, they identified both the nominal and maximum service loads and the typical time of loading. For sandwich panels, for instance, the maximum loads were listed as 70 °C and 70–10% RH for an estimated 30 days per year. These are important consideration because the envelope of VIPs is said to fail quickly above 50 °C or 85 % RH, and gas

permeation through the envelope and into the core is rapidly accelerated with temperature, which leads a slow increase in thermal conductivity, until a fail limit is attained. Yrieix et al. [142] showed that on top of these two failure mechanisms, a third one exists in the case of silica cores; on exposure to humid conditions, silica is not stable. It can dissolve, re-precipitate and sinter the effect being lower surface areas. The phenomenon is controlled by the particle size, porosity and hydrophilicity. At 80% RH, the process occurs quickly, within 20 days. In VIPs with silica core, aged at 50–70 °C and 90% RH for 400–540 days, the humid exposure translates into a 5% mass uptake due to silica. The authors suggested that VIPs be aged either for 24 h at 70 °C and 90% RH, or 30 days at 23 °C and 80% RH before thermal performance is measured, and that corresponding standard methods be updated. On the basis of the three failure mechanisms in VIPs, the authors went on to show various methods to estimate SL, a subject that is reviewed in Section 5.

### 3.4. Sealants Function and Degradation

#### 3.4.1. Function

Sealants are elastomeric products that seal openings and joints and are an inexpensive yet vital part of the building envelope. Sealants are almost exclusively polymeric in nature because of the flexibility, elasticity, adhesion capabilities and low permeability. Sealants come in a highly viscous liquid form that cures and hardens on site, this allows the sealant to conform to the shape of the opening and create a good seal. They are a crucial component to the air- and vapour-barrier system [1,29,143]. Sealants must resist shrinkage, be durable and provide adhesion to relatively unprepared surfaces [74]. Ideally they should also have a low rate of hardening and shrinking [144]. Sealant failure can result in a loss of protection for the building envelope, opening up the envelope to wind and rain damage [74]. Sealants are often viewed as a weak point in the envelope and as a result, the durability of the other components may rely on the effectiveness and durability of the sealants. They are often designed to last 20 years or more, however, there is a 50% failure rate in sealants after 10 years and a 95% failure rate after 20 years [58]. This explains why studies on the degradation of sealants are more common than any other polymeric BEMs discussed in this paper. Some of the more common sealant materials include PUs, silicones and polychloroprene [74].

#### 3.4.2. Environmental Loads onto Sealants

Sealants, like cladding, are on the first line of defence to protect structures against the vagaries of weather. Out of the loads listed in Table 3, it could be argued that resistance to UV and heat aging, along with fatigue cycling due to seasonal and day-night temperature variations, are the most important environmental loads. Heat aging, photo-oxidation, fatigue and moisture uptake are the causes of sealant aging, which can result in cracking, crazing, rupture, reduced flexibility and loss of adhesion from the substrate.

#### 3.4.3. Degradation Studies

Combined Aging Factors

In a field study, Bull and Lucas [145] reported on the 30-year outdoor Florida weathering of 13 sealants, which included nine silicones, three PUs and one acrylic terpolymer sealant. Weathering conditions had high UV radiation, RH and temperature. The goal of the study was to determine if the sealant type, chemistry and product formation would have a significant effect on how durable a sealant is and how it degrades. After 30 years of outdoor aging, sealant surfaces were examined for cracking, crazing, bubbling, discoloration and dirt pick-up. Most silicones maintained excellent surface conditions, while the PUs and the acrylic were in poor conditions. Product properties including flexibility, resiliency and toughness, were assessed by stretching, bending, twisting and gouging. Again, most silicones maintained excellent overall conditions, including good elastic recovery and low hardness, but all the PUs and the acrylic sealants had poor elastic properties. In contrast, most sealants had maintained a good adhesive bond onto the aluminium substrate.

Based on the test results, the sealants were ranked for overall durability and weathering resistance. In all cases but one, silicones ranked higher than PUs and the acrylic sealant. For each sealant type, formulation played an important role. For instance, the silicone filled with fumed silica alone ranked higher than silicones with both calcium carbonate and fumed silica. The authors concluded that sealant type, chemistry and product formulation are all important factors in the durability of sealants.

In another field study, Enomoto, Ito and Tanaka [143] examined the effect of weathering, specifically the effect of Japanese latitude, on sealants degradation. The overall degree of degradation was determined through examination of surface cracking and use of a mathematical model. Twelve samples were exposed for four years at three exposure sites in northern, central and southern Japan, where environmental conditions were monitored. Eight sealants were tested, including silicone, polyisobutylene, polysulfide and PU sealant. Their tensile strength and elongation at break were determined before and after aging. The aging resistance was based on the size and length of the surface cracks. Amongst all the products, only the degradation of one PU sealant was location dependent. The authors later expanded on the method to quantify the degradation from surface cracks [146].

In a laboratory study on sealant aging, White et al. [58] identified and ranked sealant degradation factors for their importance after statistical analysis. A systematic accelerated aging protocol was designed to provide for the quantitative effects of individually and combined environmental factor. Sealants of unknown composition were aged under different temperatures (30 °C, 40 °C, 50 °C), RHs (0%, 25%, 50%, 75%), UV radiation and cyclic movement to simulate the stresses of an active environment. Aging lasted one month, after which sealant modulus was measured to assess the effects of the aging factors. Based on the statistical analysis, the environmental factors were ranking from the most to the least significant as follows: cyclic movement, UV, temperature and RH. Cyclic movement was statistically significant for a 25% strain regardless of other factors; at 8% and 15% strain, cyclic movement was not significant, which suggests a possible threshold value for the strain. Elevated temperature was significant in that it decreased modulus, regardless of its interaction with other factors. UV radiation was statistically significant regardless of how it interacted with other factors, and RH was the least important. UV radiation repressed the effect of RH on the sample; without UV radiation, RH appeared to have a larger impact on degradation than cyclic movement. This would indicate that UV irradiance and its heating have a desiccant effect on sealants.

Sealants specifically around windows are highlighted in the next section [147].

### 3.5. Fenestration Function and Degradation

#### 3.5.1. Function

Polymers are used in fenestration primarily to insulate window frames and in glazing to reduce the thermal conductivity of window panes. Windows can occupy the largest area of a façade and can be responsible for 40% of the heat loss from a typical building envelope [148]. Agarwal and Gupta [149] have reviewed the thermal properties of plastic glazing, which are rarely used in standard residential construction. Window frames were traditionally made with wood, and then aluminium, and more recently made entirely out of polymeric materials. In these applications, polymers help to reduce thermal loss through thermal conductivity. Thermally broken window frames have an insulating polymer in the frame [148], either a PU or polystyrene foam, and polymeric frames are commonly PVC [26,148,150], although other plastics can be used [149]. Large frames for commercial applications can be made with glass-reinforced plastics [149].

#### 3.5.2. Environmental Loads onto Fenestration

Fenestration products, like cladding materials, represent the first plane of protection against weather. As such, they are directly exposed to aggressive aging loads that include UV radiation, elevated temperatures, hot-cold cycles and moisture. Windows are the thinnest portion of the building envelope, and because of large thermal gradients,

fenestration components are susceptible to moisture condensation in winter. Polymer-based fenestration products can experience surface cracking or crazing, discolouration and dullness over time when exposed to heat, photo oxidation, freeze-thaw cycles and moisture uptake [23].

### 3.5.3. Degradation Studies

UV Radiation

Gonzalez et al. [151] studied the effect of UV-visible radiation on PVC window frames. This study mainly focused on aesthetic changes on the product surface. Samples were exposed to a cyclic aging process which consisted of 18 h under UV lights at 40 °C and a RH of less than 20% for 18 h, before exposure without UV radiation at a temperature of 20 °C and increasing RH until 70% was reached for 6 h. Samples underwent this aging for up to 2000 h and then removed and tested periodically. After 500 h, micro hardness reached a minimum of approximately $10.5 \times 10^{-7} \, Nm^{-2}$ and then increased to $12 \times 10^{-7} \, Nm^{-2}$ at 1000 h and plateaued. Slight bleaching occurred in the first 1000 h, likely caused by the oxidative process on the polymer chains, while no major colour change happened over 2000 h, after which time elongation at break decreased from 50 to 30%. This type of aging is typical of chemical degradation, where an initial decrease in micro hardness could be related to polymer chain scission, and following increase in stiffness and decrease in elasticity are the result of crosslinking between unsaturation that form in PVC on aging [38,152].

Mechanical

A study performed by Pilarski and Matuana [153] aimed to determine the durability of wood flour–plastic composites exposed to accelerated freeze–thaw cycling. The wood–plastic composites studied are used in a variety of applications including window and door frames. Two composite formulations with maple and pine wood flours were tested. The wood flour was dried at 105 °C for 48 h and combined with PVC and additives such as tin stabiliser, calcium stearate, paraffin wax and others. The freeze–thaw cycling was done in accordance with ASTM D6662-01 [154]. One cycle consisted of three parts; soaking the samples until equilibrium moisture content at ambient temperatures, exposure to freezing for 24 h at −27 °C, and thawing for 24 h at 23 °C and 50% RH. The change in mechanical properties was determined by testing the flexural properties of the samples, at least eight replicates were tested to achieve an average for each formulation. The results indicated that composites with lower wood flour contents of either wood species retained their strength after cycling, the composites with higher wood flour contents experienced greater losses. The modulus of elasticity was greatly reduced for all composites regardless of wood species and content. It was also found that the maple flour composites were more affected by the freeze–thaw cycles than the pine flour composites [153].

Combined Aging Factors

In a long-term natural weathering study, Jakubowicz and Möller [155] investigated the aging mechanisms in PVC window frame aged for 20 years in Germany. Three sections were analysed: an outdoor section exposed to all environmental loads, an indoor section exposed to humidity without sunlight and interior partition shielded from moisture and sunlight. Depth profiling and analysis was carried out by infrared spectroscopy and calorimetry to determine chemical variations and impact strength. The results showed the greatest aging in the outdoor section, but degradation was mostly within 100 μm of the exterior sample surface. In this thin layer, PVC aged through dehydrochlorination to produce low molecular weight and oxidised degradation products. The chlorine released from the degradation reacted with the calcium carbonate filler to produce calcium chloride. Outdoor pollutants, such as, sulphur oxides also reacted with the PVC frame in an unknown manner. The overall weathering effect was a porous and brittle PVC outer layer with low impact strength. The inner layers were not affected by these weathering pathways, but they were not immune to aging as their impact strength was lower than the unaged PVC;

physical aging was discounted as a possible stiffening mechanism of the inner layers, and the change was attributed to an undetermined chemical process.

Fernandes et al. [147] created a methodology for the SLP of window frames with a variety of materials installed in Portugal. This methodology was based on collecting existing data and determining the general trends for the degradation mechanisms. Out of the 182 windows, 137 had polymeric sealants around the frame; 112 were around aluminium frames and 25 were around PVC frames. The aging of both the polymeric sealant and the PVC was of interest here. The age of aluminium windows ranged from 1 to 39 years, but the age of the surrounding sealant may not have been the same. The PVC windows were from 1 to 13 years old, with sealants probably the same age. Degradations or anomalies of the polymeric materials were organised into two categories: sealant around the frame, and the framing material itself with its coating. Slight sealant degradation was most common, as debris accumulated and superficial deterioration occurred in more than 86% of the window sealants; biological growth was visible in 46% of the sealant around aluminium frames and in 12% of those around PVC frames, which suggested that sealants around the aluminium frames might have been as old as the windows themselves, and at the very least, older than those around the PVC windows. This was consistent with the level of sealant debonding, which reached 43–65% around aluminium frames, and only 4% around PVC frames. With respect to the PVC window frame itself, the authors observed little degradation given the relatively short service time. About 68% of the PVC frames had an accumulation of debris on the surface, which is consistent with the work of Jakubowicz and Möller [155], but none showed the other anomalies surveyed, namely, colour change, cracking, biological growth, deformation, open joints or corrosion. Fernandes et al. [147] developed degradation curves for aluminium and wood-framed windows that had been in service up to 40 years in efforts to predict their SL, but the relatively short age of the PVC windows and the absence of degradation precluded predictions of SL. On the basis of shallow degradation after 20 years of weathering (Jakubowicz and Möller [155]), and provided effective maintenance (Fernandes et al. [147]), the SL of PVC window frames may exceed the 30 to 40 years of the wood-framed windows in the same service conditions.

### 3.6. Gaskets Function and Degradation

#### 3.6.1. Function

Gaskets are thick ribbon (tape) sealants widely used with glazing and precast concrete. They prevent leakage and thermal loss as part of the air- and vapour-barrier system [1]. Walls are always subjected to dynamic loads caused by air pressure differences, wind and mechanical ventilation, and gaskets must effectively maintain a seal under these loads. Elastic polymers are ideal materials for this application as they can stretch, compress and twist without failure [156]. Common gasket materials are ethylene propylene diene monomer (EPDM) rubber and polychloroprene (Neoprene).

#### 3.6.2. Environmental Loads onto Gaskets

The CSA S478:19 standard describes many of the degradation concerns for gaskets which include heat aging, photo-oxidation, fatigue and moisture uptake, all of which can cause cracking, crazing, rupture, reduced flexibility and loss in adhesion. Gasket degradation affect the water and air tightness of the building envelope, which can result in damage to the building. Gaskets replacement can be expensive; hence, gaskets must be durable and be designed for long SL.

#### 3.6.3. Degradation Studies
#### Combined Aging Factors

Björk and Öman [156] characterised EPDM rubber gasket used with glass coverings after 6 years of weathering on the north and south faces of a shopping centre in Stockholm. The two orientations allowed for a comparison of the effect of various solar radiation and temperature on gasket degradation. The annual mean air temperature was 6.2 °C

and the solar irradiance on the north and south sides was 330 W/m$^2$ and 640 W/m$^2$ for 4–6 h per day, respectively, from which maximum temperature loads were estimated to be in excess of 50 °C for 570 h/year on the south side and lower than 50 °C on the north side. Gaskets facing south showed higher permanent set than the ones facing north due to higher temperatures and more intense solar radiation. However, cross-linking density and elasticity measurements indicated no significant difference in the extent of degradation between the north and south facing samples. The results were consistent with the expected slow polymer chain-scission and degradation from further cross-linking below 70 °C [156]. The authors concluded that no gasket failure was expected in the foreseeable future.

## 4. Service Life Prediction (SLP)

SL, according to ISO 15686-1 [157], is 'the period of time after installation during which a facility or its component parts meet or exceed the performance requirements'. The SL of building components largely depends on the materials' properties, degradation mechanisms, environment, quality of design and work execution. As mentioned in earlier sections, the properties of BEMs change during service because of aging. This is due to chemical and physical reactions from exposure to various degradation factors such as mechanical loads, sunlight, heat, humidity and cyclic exposures to high and low temperatures, freeze-thaw and wet-dry conditions.

SLP consists of a series of tests and analyses to determine the useful life of a product when it is subjected to a variety of in-use conditions. SLP helps estimate when a product would reach a critical performance level and therefore when it would need to be replaced or repaired. This is important to both the building owners, to plan ahead for repair and maintenance schedules, and to the material or system providers to set realistic warranty periods. Moreover, decisions on product selection may be made on the basis of SLP to circumvent difficult or expensive repairs or replacements, and avoid liabilities due to failure. The more severe the ramifications of a failure, the more time and effort should be put into SLP. Traditionally SLP, and by extension, standards and policies in the construction industry are based on experience. This reactionary method works for products that have been in circulation for a long period of time. The study by Fernandes et al. [147] discussed earlier is a good example of SLP based on experience. Accelerated aging can serve as an alternative to the reactionary method by simulating conditions that cause degradation to determine protocols and preventative measures to reduce risk. Interest in SLP methods has been rising steadily since the 1960s [15,53], which can be attributed to the use of new materials with unknown failure risks. This is of great concern for the aerospace, automotive, petroleum and construction industries [53].

BEMs have properties necessary for the functionality of a building. The question of SLP is how do different aging mechanisms alter these properties and at what point, do these properties no longer meet the requirements for a functional and safe building? To provide with an example, a review paper by Bomberg, et al. [158] discussed the limitations of current laboratory test to determine the moisture resistance of weather resistive barriers (or vapour barriers) in the building envelope and the newer methodologies developed. They concluded that current tests are largely good for quality assurance, but due to a number of factors, they do not provide precise enough information to evaluate barrier performance in wall assemblies. The goal then would be to develop more comprehensive tests that yield more useful information. SLP aims to take this further and seeks to define the performance limits for material properties affected by both installation and in-service aging to ensure confidence in the long-term use of a product or component. The ASTM C1850-17 [19] outlines a procedure to develop SL methods for sealants, and it discusses the importance of understanding the limitations and requirements of a material, and also of SLP methods to help define stages in material degradation. This is a challenging process, as information on material properties are usually provided by manufacturers, and they can be incomplete, biased and inconsistent from one product or producer to the other. The implementation of SLP would help to remedy this situation.

Service conditions directly impact on the way that materials degrade and by extension of the useful SL. This makes geographic location a significant factor to determine the SL of a product. Therefore, SLP needs to be carried out on a component to component basis, and there is no set standard for all buildings, so no two buildings are exactly alike in composition, material, design or location [22]. To determine the SL of a building element, the SL of the structure, the structure environment, the quality management procedures, the inspections, maintenance and repair schedules, the consequence of failure and the expected availability of repair components, must be considered [23].

### 4.1. SLP Methods

When a new product is developed for the construction market, no performance history exists. This is where SLP is most useful. The fundamental steps of SLP, no matter which method is being applied, remains the same [49]. The first step is to define failure, so that acceptable degradation levels are quantified. The second step is to gather information on service conditions of the material; including the type, magnitude and duration of each significant source of degradation. In this step, it is critical to consider all the possible sources of degradation, such as long-term duty cycles and extremes. The third step is to decide on either an experimental or a modelling approach. In the modelling approach, fundamental degradation mechanisms and kinetics are considered to arrive at a potential SLP model, and then the model is validated with experimental work. The experimental approach entails the study of material response to degradation factors either directly in the field or in the laboratory. In the latter case, climatic loads that simulate in-service conditions are applied through the use of accelerated aging techniques.

Laboratory aging is most common because it is cheaper and faster than field aging. In accelerated aging, the degradation loads are brought outside of their natural range to reduce the time to material response. The key in accelerated aging is to ensure that simulated lab conditions are representative of the service conditions, in other words, the degradation process must be accelerated while the degradation mechanism remains close to the natural field process. In field aging, an alternative approach to accelerated aging in the laboratory, samples are maintained in a natural environment to be examined over time; these studies can be carried over a few years to validate the laboratory-aging mechanisms, and sometimes they last for decades.

SLP methods are often most accurate and useful when used in conjunction. A model can guide the setup of experimental work and help validate experimental findings or, vice versa, experimental results can be used to create SLP models. Next, the experimental and modelling approaches are discussed in more detail.

### 4.2. Experimental Approach to SLP

4.2.1. Field Aging

Weathering in the field, or natural aging, to probe in-service performance is the better way to assess the degradation of BEMs; that is to say, it provides for true aging pathways and mechanisms. However, weathering is generally too slow to provide timely results for the selection of innovative materials, as part of a new construction project, for instance. Field aging may be used best to further the knowledge of products with some history, and this may be most useful for products intended for mass markets, and to help document typical long-term performance for a specific geographic location. This is because field aging provide for unique and non-repeatable aging loads that hardly transfer from one location to another. For instance, tests for the same cladding exposed to weathering in Montreal, Houston, Buenos Aires, Paris, Shanghai or Brisbane would not provide for the same results as climate conditions will vary greatly [14]. Consequently, to fully understand the material degradation pathways, multiple tests need to be run in a variety of environments [34].

One of the greatest benefits of field aging is in the production of data related to climatic loads, that is to say, the focus is not on the control of weathering loads but on the monitoring of exposure factors [14]. Field aging tests must be designed to provide accurate data under

specific environments [34,110] and sample attachment to some framework must simulate in-use conditions and strains [34,84]. As daily weather is not constant, longer exposure periods provide for a more accurate average and limiting loads [34].

There are three basic approaches to gather data from field tests: (a) Field exposure sites, either in an open field or on a rooftop, for direct exposure to sunlight; (b) experimental buildings; and (c) building inspections. The approach is selected based on the sought data, for instance, solar irradiance, RH, pollutant concentration, rain level and its pH and the type of product tested. In this respect, for example, a vapour barrier would be best tested with approach (b), and not (a), because these barriers will generally not be exposed to UV light outside of the construction period, and approach (a) would introduce unnecessary UV loads with possibly substantial material degradation not representative of true in-service aging [24].

Field Exposure Tests

In field exposure tests, samples are directly exposed to an exterior environment for an extended time. The test location usually brings an environment with a specific degradation factor, a sunny climate to test UV degradation, a location with sub-zero winters to test freeze-thaw degradation, a rainy environment to test moisture degradation [34,84].

Kimberlain et al. [159] performed reliability checks to compare the real and predicted SL of sealants. They evaluated the performance of silicone sealants, including adhesion and elasticity, after 55 years of service [159]. Their study showed that the sealant could reasonably retain essential elastic properties after exposure to extreme temperatures, UV radiation and wind-driven rain for more than five decades. In another study, Ito et al. [160] performed in situ weathering test to evaluate several different types of construction sealant for a period of 10 years. The sealants were weathered at three sites with different climatic conditions, and under imposed cyclic movements. The waterproofing function of the sealants was thereafter evaluated from cracks in cohesive or adhesive failure. They concluded that sealants located in cold climates showed higher failure rate than sealants in subtropical climates.

Experimental Buildings

Experimental buildings are constructed specifically to test materials and products. These buildings range from simple huts to test the resistance to direct weathering loads, to complex buildings where the structure may have interchangeable walls and roofs, or simulate family life in the interior where the shower will run at pre-set intervals, lights turn on and off, heating systems run in winter and air conditioning runs in summer, to test the internal and external impacts [34]. For instance, Möller et al. [110] studied LDPE barriers in a small dismountable structure to assess long-term aging; see Section 3.2.3 for results. At the other end of the spectrum, Riahinezhad et al. [31] reported temperatures and moisture conditions in the envelope of a test house with full services, and where the actions of a family of four persons was simulated.

Building Inspections

Building inspections planned or ad hoc examinations to track the changes of a BEM or the building as a whole. Most inspections that include materials assessment are descriptive in nature [34]. They are most often test cases, but they can highlight general trends [161]. It is difficult to determine changes in physical properties of materials with inspections, as testing often requires a sample to be retrieved for off-site analysis, leaving an area of potential weakness in the envelope. An in situ non-destructive testing protocol is a key to monitor the condition of BEMs during inspections. Fernandes et al. [147] and H. Orr et al. [112] both completed degradation studies based on building inspections, which were discussed in earlier sections.

### 4.2.2. Laboratory Aging

Laboratory aging or laboratory weathering began in the early 20th century [49] and it is often designed to mimic the conditions a material will undergo while in service [14,49]. Laboratory exposure must be designed to control independent variables precisely and accurately [14]. Multiple methods may be used to study the aging process in a reasonable amount of time [24], and accelerated aging factors may be applied either in series or in parallel [68] as exposure to various levels of UV, humidity and temperature simulates the service environment [32].

Accelerated laboratory aging can provide rapid information on material degradation, but improper experimental design may provide misleading results as failure modes may not be representative of failures in the field [22,34]. As an example, standard method ASTM G154 has been used to age sheathing membranes under UV light [45]. This method is inexpensive to conduct because the instrument is simple and the fluorescent UV lamps are cheap to buy. However, the conditions in ASTM G154 are not representative of sunlight as the UVA-340 lamps used in the method only emit light under 400 nm, leaving out the sunlight-related degradation effects of visible light and infrared radiation [37]. Polymer products with UV stabilisers have been known to pass ASTM G154 and fail in the field. Pure polyolefins degrade almost exclusively from radiation under 340 nm, in which case testing as per ASTM G154 may be warranted, but in practice water sheathing membranes are not fabricated with polyolefins alone, with pigments like titanium dioxide that can significantly modify the performance of exposed materials [37]. In consequence, a conservative approach to UV aging relies on ASTM G155 [46], for which a Xenon-arc is used to produce UV, visible and infrared light close to the entire sunlight spectrum. With Xenon arcs, daylight filters must be used; historically, Type-S borosilicate inner and outer filters have been used on Atlas Weather-O-meters, and Daylight-Q filters have been used on Q-LAB instruments. Recent procedures, like those in ASTM D7869-17 [162] require filters with a better match with solar irradiance in the range of 340 nm to 400 nm. This translates into the use of a Rightlight filter on the Atlas Weather-O-meter, and a Daylight-F filter on Q-LAB instruments.

SLP methods are difficult to design because many factors contribute to material degradation; a simple list of degradation factors is next to impossible to write [22]. Knowledge of fundamental processes involved in material degradation is essential to develop reliable accelerated aging methods, and yet these methods only consider the most significant aging loads to simulate specific climatic conditions. For example, cladding is exposed to UV light in laboratory tests, but not to alkali, although they may be in contact with alkaline concrete during service [34,116,163].

The reliability of laboratory tests is best validated by comparison with field tests. Ideally, this requires field samples aged under noted conditions and retrieved at different time intervals, laboratory samples aged under accelerated conditions, also retrieved at different time intervals, the laboratory analysis of both field and laboratory samples to determine degradation rates and mechanisms and the comparison of results from both aging regimes. High correlations between field and laboratory results validate the accelerated aging approach [49].

Considerations for Accelerated Aging Tests

Accelerated aging of polymeric materials have been discussed and practiced for almost a century. Notwithstanding, myths and misinformation remain as roadblocks in accurate durability assessment and SLP. These issues are noteworthy:

- The term SLP may be used without recognition for the complexity of this task. Pickett et al. [49] explained the challenges before reliable SLP and the need for a multidisciplinary approach. For the building envelope illustrated in Figure 1, consider the many variables and factors that influence the SL of each element; SLP cannot be achieved without a thorough analysis of the system components and their interaction, and the combination of expertise is an asset for accurate SLP.

- SLP from accelerated aging without detailed knowledge of a material's service condition is not possible. Actual service conditions must be defined or measured to design appropriate accelerated aging conditions [31].
- Accelerated aging does not mean large increases to aging loads. Each increase must be justified. As an example, very intense UV irradiation in accelerated aging can lead to an aging mechanism not representative of reality [164]. At high UV radiation, the surface of the polymer goes through rapid photo-oxidation, which results in diffusion-limited oxidation atypical of the natural process [49]. Likewise, accelerated aging at temperatures beyond a polymer phase transition (e.g., glass transition temperature) leads to greater than normal oxygen diffusion rates and degradation in the polymer phase that remains glassy at service temperatures, which leads to degradation mechanisms and activation energies different from that in normal service temperatures [16,135]. Therefore, aging loads should be raised for a reason. When limited background information is available, short preliminary tests to determine the initial material response to a high aging load can help to plan an appropriate accelerated aging test.
- Prescriptive aging standards developed for quality control can be a barrier to advances in durability testing and SLP [15], which require performance-based standards [165].
- A single accelerated aging protocol cannot simulate in-service conditions for every material, or climate. Aging protocols must be tailored to both service conditions and material properties. In other words, one accelerated testing protocol might work very well for a class of material, while it fails for another one [37]. As a result, experience with similar products or materials can provide insight on degradation pathways and a template for new tests, but it cannot replace the SLP for a similar product because slight variations in product chemistry can alter aging resistance [17].
- Accelerated aging and SLP is a complex endeavour. Knowledge of organic chemistry and polymer physics can help to understand degradation mechanisms and to develop accelerated aging methods. Knowledge of statistics and mathematical models helps describe product aging and extrapolate to possible failure times [17,56].
- Mathematical models may be extremely useful for SLP, but they must be used with care, as ample information and verification is required to achieve predictions with high confidence levels [17].

Validation of Laboratory Aging

The results of accelerated laboratory aging are validated when they highly correlate with results from field aging, this includes the aging mechanism(s). Once the results of accelerated aging are validated, then mathematical models can be pressed against the data to obtain an accurate SLP model. Poor knowledge of failure modes, and a lack of methods to quantify the effects of aging factors, either in laboratory or field samples, can lead to poor correlation between results of accelerated aging and field tests [58]. In this section, representative studies with successfully implementation of SLP will be discussed.

Hardcastle [18] developed a simple, yet effective SLP model for polycarbonate in window glazing [149]. The work was based on the correlation of a yellowness index (YI) for samples aged in the laboratory and in the field. Field samples were placed at outdoor exposure sites in Miami and Phoenix, where UV intensity, black panel temperature and RH% were periodically measured for over 3 years. Based on these measurements, 'an average' climatic year was defined. In addition to the climatic conditions, the YI was measured and failure was defined as YI = 6, or greater. In this study, the author related the time to failure in the field to simple cumulative UV irradiance. The time to polycarbonate failure with YI = 6 was observed when UV radiant exposure at 295–385 nm reached 380 and 480 MJ/m$^2$ in Miami and Phoenix, respectively. In the laboratory, polycarbonate was aged under the UV light of a xenon arc and the light irradiance was varied within the range measured in the field. The YI and time to failure was measured. Based on this data, an Arrhenius-type model was developed, where time to failure was related to irradiance.

To validate the predictive Arrhenius model obtained from laboratory data, a new time to failure was calculated but with the UV irradiance in the field from the 'average year'. The predicted failure time from the model showed reasonable agreement to the actual time to failure in outdoor aging.

This study by Hardcastle [18] highlights some key points to build a successful SLP model. First, good correlation between the laboratory and field aging mechanism provided the basis for a reliable prediction model validated with actual time to failure in outdoors testing. Second, assumptions can be verified with a comparison of field and laboratory data; in the case, the assumption was that yellowness arose from UV irradiance and not from the other aging loads. Third, field aging requires several years to capture natural environmental variations and calculate climatic loads for an average year.

Gu et al. [166] performed SLP on an epoxy used in common in construction adhesives and concrete repair materials through field and accelerated aging. The epoxy was a combination of diglycidyl ether of bisphenol A and 1,3-bis(aminomethyl)-cyclohexane. Field exposure consisted of 20 months of outdoor aging over 4 years on a south-facing roof. The temperature and RH were recorded regularly. Samples were analyzed with FTIR and UV-visible spectroscopy every 3 to 4 days until failure was reached. Parallel laboratory tests were simulated using the NIST SPHERE (simulated photo-degradation via high energy radiant exposure). Tests were run at four temperatures, four RHs and four wavelengths, each at four spectral intensities. Samples were also analysed using FTIR and UV-visible spectroscopy every 3 to 4 days. The effect of aging on the samples was observed by comparing the total effective dosage, defined as the radiation absorbed by the sample, with the change in absorbance of significant IR bands determined by FTIR. Dosage was used instead of time in linking exposure to degradation, meaning that each sample would require a different length of time to reach the same level of degradation due to differences in conditions, but total effective dosage could be determined and linked to degradation regardless of the length of the exposure time.

Linking the field and laboratory exposure results is the key to validate SLP models. This study utilised three techniques to demonstrate how a link can be determined. The first method used ratios of IR bands absorbance to determine if degradation mechanisms present in both aging tests were the same. To achieve this, two band-absorbance readings from the same sample were plotted against each other. The plots revealed a similar trend for each sample indicating the accelerated aging method accurately simulated the degradation method.

The second method used a computer program to estimate the damage of outdoor samples based on the damage/dosage curves from the SPHERE samples. The temperature, UV radiation and RH of the outdoor tests were inputs to the program and the damage/dosage curves and conditions for the SPHERE tests were treated as 'calibration curves', to estimate the overall damage. Estimates were performed for one sample and were compared to the results from outdoor aging. The trends matched very closely and demonstrated how material degradation could be estimated with laboratory tests.

The third method was a prediction model based on the total effective dosage model and the accumulated damage developed by I. Vaca-Trigo and W. Q. Meeker [167]. The parameters for temperature, RH, wavelength and light intensity in the model were based on the SPHERE results. The calculated damage was compared to the observed outdoor tests, the data followed the same general trend but was not as accurate as the second method of estimation, because parameters were based on the estimation rather than a function of the environment. However, this method still provided a general demonstration of material degradation upon exposure to specific aging conditions. The work done by Gu et al. [166] demonstrates how to successfully relate accelerated aging data to field exposure and these concepts could easily be applied to SLP.

Y. Lv et al. [168] correlated the laboratory and outdoor aging of PP, a polymer common in air barriers and cladding. Outdoor weathering was at six stations in China, each with a different climate. Weather data for each sites were logged daily. Samples were aged up to

two years, and they were collected after 30, 90, 180, 360, 540 and 720 exposure days. In the laboratory, accelerated aging followed the ISO 4892-2 procedure [169], with temperatures ranging from 55 to 70 °C and RH of 65 to 100%, continuous UV irradiation, and wet/dry cycles. Samples were collected every 3 days, for 30 days.

Outdoor weathering resulted in polymer oxidation in all climates, as measured by the infrared spectroscopy. Degradation occurred by thermal- and photo- oxidation and depended greatly on the location, with greater degradation was measured in samples from warm and humid climates due to higher temperatures, oxygen pressure and annual irradiation. This trend from spectroscopy was consistent with the results seen from chromatography (molecular weight reduction and dispersion) and scanning electron microscopy (presence of cracks). In the laboratory, accelerated aging resulted in oxidation measured by infrared spectroscopy and was consistent with field aging. Similar to the outdoor aged samples, laboratory-aged samples showed lower average molecular weights and greater polydispersity. In general, the results showed similar trends from both field and laboratory-accelerated aging.

On the basis of the results from physico-chemical analysis, a degradation model was developed; the mathematical relationship was a modified Arrhenius equation that included three aging factors, namely, the UV irradiance level, the oxygen pressure and the standard temperature. A linear relationship between oxygen and degradation rate was assumed, which was consistent with the literature review of Vink and Fontijn [170]. From the ratio of data from the laboratory and the field, an acceleration factor was calculated. The laboratory aging provided for the greatest acceleration over the cold climate, and the lowest acceleration over the warm and humid climate, with respective acceleration factors of 30 and 8. With the acceleration factors, the evolution of the oxidation and weight average molecular weight in laboratory samples could approximate the results from outdoor aging.

The work of Hardcastle [18] on polycarbonate, Gu et al. on epoxy [166], Lv et al. [168] on PP are case studies during which laboratory-accelerated aging was compared and validated on the basis of field aging. In all cases, the materials were exposed to sunlight, which would be the case in construction for cladding or roofing products, but not for other BEMs, and in no time were the case studies conducted on actual BEM. We could not find studies on actual BEMs where laboratory aging is compared to field weathering.

### 4.3. Modeling Approach to SLP

A most important step in SLP is the use of experimental results in the creation, or adjustment, of numerical models that describe material aging. There are two primary types of models that may be used for SLP, they are empirical or theoretical [17]. Empirical models are developed from statistical analysis of data collected during accelerated laboratory aging and field aging tests. These models may fit the acquired data very well, but caution must be used in extrapolation with these models because there may be no physical meaning to the mathematical trend. This is the case for instance with Equation (1), where A and C are constants, and which may be used to model the effect of temperature $T$ on the reaction rate $k$. This equation has no physical basis [171].

$$k = A \times T \exp(C) \tag{1}$$

An empirical model is best used for extrapolation when it is supported with knowledge of the underlying degradation mechanism(s) and enough experience to promote confidence in the results [17]. Theoretical or semi-empirical models based on physical and/or chemical phenomena can provide greater credence to extrapolations and predictions. As an example, the Arrhenius model (Equation (2)) relates reaction rate k to reciprocal absolute temperature 1/T, the gas constant $R$ and an activation energy $E_a$, [172]. This initially purely empirical model finds theoretical grounds and can be stated in thermodynamic terms, where the activation energy is related to the reaction enthalpy and the

pre-exponential factor *A* is related to the reaction entropy [54]. The Arrhenius rate model thus classifies as a semi-empirical or semi-theoretical model.

$$k = A \times \exp(-E_a/RT) \tag{2}$$

The Arrhenius model is most common in thermo-oxidative-accelerated aging tests where short-term exposure at some elevated temperatures is related to longer times in service temperatures. A log plot or inverse plot ($1/k$ vs $1/T$) for a simple reaction would then present a linear relationship. Meeker and Escobar [17] accounted the Arrhenius equation to define an acceleration factor (AF) for a thermally driven aging process as per Equation (3), where $T_s$ is the in service temperature, $T$ is the higher aging temperature. This AF is the ratio of the accelerated reaction rate divided by the in-service reaction rate, which for building envelope materials would be the reaction rate at the average in-service temperature, e.g., 15 °C [31].

$$AF = \exp[E_a(1/RT_s - 1/RT)] \tag{3}$$

Meeker and Escobar discussed some examples where the results did not correspond to the linearity of the Arrhenius equation, the possibility of a secondary reaction overtaking the first at higher temperatures was used to explain this phenomenon [17]. These authors discussed other considerations in the analysis of accelerated aging data, including the use of approximate $E_a$ from similar materials versus its expensive and labour-intensive experimental determination, and the impact of the approximation on the overall accuracy of the results.

The Arrhenius equation applies to single reactions of the type A + B = C that occurs in one step [171]. The degradation mechanisms for polymeric materials are often complex, thus the use of the Arrhenius equation for polymer degradation can be complicated, and measured activation energies may be averages of several reactions. In a review paper, Celina et al. [173] discussed the uncertainties with the application of the Arrhenius model to polymers and highlighted cases in which a linear relationship was not observed; often non-linearity occurs over a threshold temperatures [17], or to the limited availability of oxygen for reaction in the bulk (diffusion limited oxidation). In these cases, the model may be adjusted with two $E_a$ values, one below and one above the threshold. Other effects may also limit the use of the Arrhenius model, including physical changes, crystallisation effects and changes in stabiliser solubility. For complex cases, Celina et al. demonstrated how the Arrhenius equation may be adjusted to better represent systems with multiple reactions; each reaction has a rate constant ($k_1$ and $k_2$), adding these together gives an overall rate constant ($k_{sum}$). For the case of two competing reactions, the adjusted equation then consists in two terms, one for each reaction. Each reaction gives its own activation energy and pre-exponential constant as shown in Equation (4).

$$k_{sum} = k_1 + k_2 = A_1 \times \exp(-E_{a1}/RT) + A_2 \times \exp(-E_{a2}/RT) \tag{4}$$

This equation better represented data with curvature than the basic Arrhenius equation [173]. In this case, linear extrapolation can no longer be used as the relationship is no longer linear, but an acceleration factor could still be defined from the duality of Equation (4).

The SLP models highlighted in this section are good examples of models with a physical or chemical foundation from which accelerated aging tests and lifetime prediction may be made with confidence. The scope was limited to the standard Arrhenius model for simple reactions, and its extension to competing reactions, but there are other models with physico-chemical foundations, including the Eyring model [57].

## 5. Significance of Research on Durability Assessment and SLP

SLP would be a useful tool to assess the durability of BEMs. Products could be compared, not on the basis of unaged properties, but under service-like conditions that

account for climate change, more accurate life cycle cost analysis could be conducted, repair and maintenance schedules could be set for a building or its envelope as a whole, and more durable material could be used to reduce demolition waste for environmental benefits. This section highlights the significance of conducting research on durability assessment and SLP.

### 5.1. Modeling Approach to SLP

Climate change arises from the sum of changes to the atmosphere, oceans, chemosphere cryosphere, biosphere and land hydrology [174–176]. The changes to the earth systems are caused by anthropogenic forcing agents, which include greenhouse gases, sulphates, black carbon, organic carbon and industrial dust [174,177,178]. Climate change simulations by models project the current trend over the next 50 to 100 years and predict possible climate variations such as mean precipitation and frequency of extreme. For example, the National Building Code of Canada [20] and standard CAN/CSA S6.1-14, a commentary on CSA S6-14—Canadian Highway Bridge Design Code [179], projected changes in environment based on a 2–3.5 °C increase in global average temperatures. Based on a 2 °C global increase, an increase between 4 and 16 °C is projected across Canada for January, and 1–9 °C for July. Temperatures are projected to be much higher across Canada in the case of 3.5 °C increase globally, and a significant increase in precipitation is projected (up to 50%), particularly in northern Canada [20,179]. Climate change also means greater aging due to prolonged periods of higher temperatures, higher levels of UV radiation and higher RH due to increased wind-driven rain [43,178]. In 2017, global warming had caused an increase of 1 °C globally above pre-industrial levels, increasing at 0.2 °C every decade [180]. It is believed that without government policies and interventions, the global increase in temperature by 2100 will be much greater [180,181]. In terms of SL of building materials, an increase in length and severity of heat waves, increased rainfall or extreme weather events such as hurricanes will directly impact the rate of degradation and the service life of BEMs.

Durability designs and aging protocols are based on experience, and current standards for BEMs are at best to ensure acceptable SL under historical and non-warming climates. In other words, current durability protocol will hardly help to predict performance and failures under future climatic loads. There is a big knowledge gap in the expected performance of BEMs under future climatic loads. Climate change may significantly alter historical design considerations because of the rise in the frequency and amplitude of climatic loads and the unknown ability of BEMs to resist these new loads. Longer exposure times to more aggressive climatic loads may cause more rapid deterioration of materials and systems, and thus reduce the resistance to aging and shorten the SL. A reduced SL from polymeric BEMs would have an impact on the SL of structural materials and components, which would raise concerns over the structural performance of any buildings, and affect the health and safety of building occupants. Therefore, it is critical to consider today the possible effects of future climate change on the durability of BEMs and assessments of SLP.

### 5.2. Improving Life Cycle Cost Analysis

SLP has an important role in the economic and environmental sustainability of the construction sector. The knowledge of the SL of BEMs is necessary in economic analyses, such as life-cycle costing (LCC), and environmental analyses, such as life-cycle assessment (LCA). The LCC analysis is a way to estimate the cost of a building construction and maintenance across its entire SL [23,182]. New energy-efficient and durable materials can lead to a longer SL for buildings; thus helping in reducing the life time costs of the building [10], which is determined by the total cost of acquiring, owning and disposing of the building [172–174]. It is expected that as better standards for materials and energy efficient buildings enter the market, the LCC will continue to be reduced along with the operation costs [10]. LCC includes the initial costs, along with the operations, demolition and repair and replacement costs. Polymeric BEMs, in general, are less expensive to

purchase and transport than traditional building materials, which reduce construction costs. SLP and knowledge of the degradation mechanisms in polymeric materials will help to perform accurate LCC analysis, and may even lengthen product SL, thus reducing replacement and repair costs.

SL of building components and elements, including polymeric BEMs, is an extremely important input in LCA tools [183]. The degradation of BEMs occurs due to natural aging and leads to lower energy efficiency and higher LCC [36]. In fact, SL is proved to be the most influential factor in a building LCA. As a result, standard values of BEM SL in SLP, instead of actual values would introduce significant uncertainty and error in the analysis. The lack of proper durability and SL data also results in non-representative LCC and LCA assessments. The estimated SL of BEMs are currently based on the ISO 15686 method. However, it is clearly preferable to predict performance based on laboratory or real-life experience and data, where microclimate is well-known and the performance of the building component or element is accurately characterised. The cost of inaccurate SL values and non-representative LCC assessments are borne by building owners through the initial building cost and the increased maintenance or refurbishment costs. Without a proper plan, the building operation cost may be more than budgeted because of unforeseen maintenance and refurbishment.

### 5.3. Prevention of Premature Failure in Polymeric BEMs

The current building industry focus on durability is in part a reaction to the perceived lack of it. Warranty claims and call-backs related to polymeric BEMs are increasingly leading to a rise in litigation and insurance costs [3]. On the other hand, the management of BEM maintenance becomes increasingly important. It had been found that some countries applied 40% of their building resources to the repair and maintenance of existing buildings as a consequence of the lack of resistance to deterioration of buildings. The high costs associated with premature failure, the progressive degradation of polymeric BEMs and their maintenance and repair have an important role. This could lead to an early demolition of the building whereas more durable materials could address this problem. Durable building design is also linked with economic considerations. To give an example, despite being inexpensive building materials, sealants are an important factor in the LCC of a building. Sealant failure can result in a loss of moisture and weather barrier properties for the building envelope, a loss of energy efficiency and even building envelope materials deterioration. To avoid the costly consequences of sealant failure and serious damage to the structural integrity of buildings, it is important to understand and predict the SL expectancy of sealants. Therefore, knowledge of degradation pathways can help to improve overall structural durability and implement policies or guidelines to help prevent failure, and plan to repair or replace components before expensive failures occur. Currently, there are no consistent data on durability issues and premature failure of polymeric BEMs in service.

### 5.4. Reduction of Construction and Demolition Waste

Up to 40% of all waste can be attributed to construction and demolition of structures, particularly in countries with rapid urbanisation and low landfill fees [150]. The cost of separating end-of-life plastics from certain waste streams is prohibitive, especially when compared to other available management options (e.g., low landfill tipping fees). As such, the construction sector will likely play an increasingly important role in the overall performance of plastics value recovery in Canada [184], where the construction industry has produced large amounts of waste for decades and the volume increases continuously in many regions [150]. SLP would help to reduce the waste produced through greater structural durability and use of recyclable materials. Construction is the second largest market for plastics after packaging, with 26% of the end-use market in Canada [184]. Plastics typically represent only 1% by weight of the total construction and demolition waste, which may seem like a relatively small amount, but their long degradation time translated into significant long-term environment impacts. While the construction sector is still

a relatively small plastic waste generator, its share progressively increases. Given the rapid incorporation of plastics in construction in the 1980s and 1990s, which now remain 'stocked' in buildings, construction plastic waste will change in future years with construction renewal and as plastics from the construction sector enter the waste stream [184]. The most common plastics in construction today are PVC, HDPE, LDPE, EPS, XPS and PU [150]. Due to the small contribution to total construction and demolition waste, recycling efforts for construction plastics have been focused on the largest contributor, namely PVC, used for piping, window and door frames [150]. Greater knowledge of polymeric BEMs failure mechanisms, reliable durability assessment and SLP would help to address other polymers than PVC, and more importantly, it would help to reduce the entire construction waste stream from plastics [184].

## 6. Knowledge Gaps

A beneficial outcome of the above review on the degradation, durability and SLP of BEMs is in the identification of knowledge gaps in the field of BEM performance, and especially with respect to SLP in evolving service conditions caused by climate change. The literature review has shown that technical approaches exist to assess SL, but no studies could be found on the SLP of actual BEMs, either exposed to simulated historical weather loads, or projected loads, except for building sealants. Future work on BEMs other than sealants should therefore be focused on the development of SLP methods analogous to that used for sealants, namely, an approach based on ASTM C1850-17 [19], if this standard does not apply as is to BEMs.

Durability requirements for BEMs also need to be clarified. For instance, is there a national standard that provides specifications? In Canada for instance, the durability requirement for buildings and components is established by standard CAN/CSA S478-19 (Table 5). The minimum design service life (DSL) for buildings can be short, up to 10 years for minor buildings, and up to 100 years or more for monuments and heritage buildings. The minimum DSL of 25 and 50 years applies to the vast majority of the building volume, which is the scope of this review. A simple comparison between the building DSL (Table 5) and the assumed lifespan of some plastics (Table 1) highlights the gap difference that may exist between the sought and the actual durability of polymeric BEMs. This difference helps to explain the current need for regular building maintenance, including product replacement or repairs of readily accessible polymeric building components, which are directly exposed to the aging elements, e.g., roofing membranes, composite doors and windows. It is also noteworthy that specific product types in the building envelope (Figure 1), have no documented reference service live values (RSLV). In other words, no durability period has been established or SLP has been performed. Indeed, databases that include the RSLV for building elements, generally cover interior elements like floors, walls and finishes, roof coverings, exterior insulation, doors, windows and structural elements [185–188], but not the out-of-sight BEMs. Yet, standard CAN/CSA S478-19 requires BEMs to last as long as the design life of the building itself because BEMs are concealed and not readily accessible for repairs [23], which may be prohibitively expensive if carried out. In addition, the risk exists that materials that are hidden from sight put at risk the entire building structures, and ultimately shortening the building life and raising the volume of demolition waste and the environmental cost, not to mention dramatically reducing the value of assets. One also needs to consider that the climate change can only exacerbate the risk. Generally, a gap may exist between BEM performance requirements and established RSLV. There need to be a RSLV for BEMs in standards or in the building code. The Dutch experience may be one to follow. Already in 1995, the Netherlands established the RSLV for 600 building products on the basis of historical performance as judged by experienced building professionals [188].

Two approaches may be followed to provide a database of RSLV for BEMs (a) that are based on historical performance as reported by building professionals, e.g., the Dutch approach [188], and (b) the development of a database based on experimental work that

involves accelerated aging, e.g., ASTM C1850-17 [19]. In light of climate change, when historical weather patterns and tendencies no longer hold, the second approach may be more appropriate, but it would still lack predictive power because of unknown future climatic conditions, which must be inputs for accelerated aging. Hence, climatic projection modelling is necessary with either (a) or (b).

**Table 5.** Categories of DSL for buildings adapted from Reprinted with permission from Ref. [23]; Copyright 2019, Canadian Standards Association.

| DSL Category | Building Type | Minimum DSL for Building, Years | Range of DSL, Years |
|---|---|---|---|
| Short life | • Bunkhouses, sales offices<br>• Minor storage buildings | - | Up to 10 |
| Medium life | • Low-hazard industrial<br>• Temporary Buildings | 10 | 10 to 25 |
| | • Mercantile<br>• Medium-hazard industrial<br>• Business and personal services occupancies<br>• School portables | 25 | 25 to 50 |
| | • Low-rise commercial and office buildings<br>• High-hazard industrial | 25 | 25 to 99 |
| Long life | • Single-unit residential<br>• Multi-unit residential<br>• Mid- and high-rise commercial and office buildings<br>• Post-disaster buildings (e.g., hospitals, power generating stations, public water treatment facilities and emergency response facilities)<br>• Performing arts buildings, arenas, schools and colleges and other assembly occupancies<br>• Detention, care and treatment occupancy | 50 | 50 to 99 |
| Permanent | • Monumental and heritage buildings | 100 | 100 to 300 |

In Canada specifically, the development of a RSLV database for BEMs would follow the precepts of standard CAN/CSA S478-19 (Figure 4), and help to answer the question on whether or not the SL is in excess of 25 or 50 years, as per medium- and long-life buildings (Table 5). The steps to acceptable SL include: (a) historical environmental load assessment and future climate predictions, (b) assessments of degradation rates and mechanisms and (c) SLP modelling to establish a RSLV that account for the expected climate change. The first element is obtained through gathering historical environmental loads and simulations for future climate. Some climatic predictions for Canada up to year 2100 have already been published [189,190]. In conjunction with the current building envelope temperature and humidity loads, these predictions could provide the basis for heat and moisture transfer modelling through the building envelope to determine the expected service conditions for the various layers of the building envelope. The second step on degradation is covered in Sections 2 and 3 of this paper. The third step on modelling was reviewed in Section 4.

With respect to SLP, laboratory-accelerated aging methods that can simulate both current and projected climatic loads are required. The validation of these methods must first come from comparison with data for materials aged in the field, and a possible approach here is to retrieve and analyse materials from buildings that are being repaired, with the attempt to reconstruct product history, rather than starting new field work into a climate changing scenario and its associated uncertainties. The validation of laboratory-aging conditions with the increased climatic loads predicted for the future will likely be a modelling effort, with the necessary step of model validation based on the 'prediction' of historical field data. On a successful outcome, the model may then be extended to future conditions, that is, to account for future expected multiple loads. Possible models may be extensions of the Arrhenius-like model in Equation (4) with a rate constant for each of the aging loads from Table 3. The forward path is not likely to be straight, and significant efforts will be required to complete these tasks and address the multiple knowledge gaps.

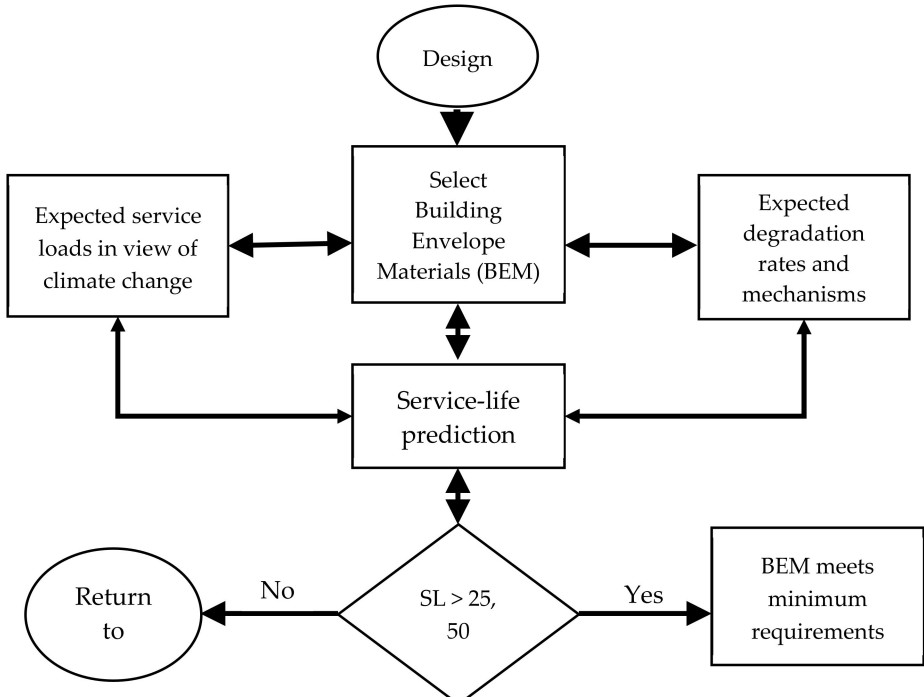

**Figure 4.** Steps to the selection of BEMs with SL greater than 25, 50 years.

After SLP work, and after the associated databases of RSLVs have been established, such values must be referenced by building codes, regulations and standards. Durability studies and SLP are relatively expensive and long to undertake, and in Canada only recently has there been impetus towards the needs for SLP in the building code. For instance, the former guidelines for durability in building, CAN/CSA-S478-95, are referenced for information only in Annex A-5.1.4.2 of the 2015 National Building Code of Canada [20]. However, the former guidelines have been updated to a full standard under CAN/CSA-S478-19 [23] and formal reference to this new durability standard may be expected in the body of the Code in the near future. Such references to durability in building codes and regulations will provide further impulse to advance efforts on SLP of BEMs, and associated RSLV databases.

## 7. Conclusions

The goal of this critical review on the degradation and durability of polymer-based BEMs was to identify knowledge gaps in the SLP of BEMs with a perspective on climate change and its possible effect on BEM durability. We found extensive evidence for the degradation of the polymers used in the building envelope. This includes for instance PE,

PP, PS, PU and PVC, and studies provide indications of degradation pathways. In very rare instances, however, the work pursued to the extent that SLP of actual BEM can be made, either in the laboratory or the field. The only exception is the case of silicone sealants, for which degradation has been systematically studied, and from which a SLP framework has been formalised, i.e., ASTM C1850-17. In Europe, SLPs for construction products are collected as RSLVs in national databases, most often based on expert opinion, but this approach lacks any provision for the future performance in a changing climate. The minimum SL of BEMs is either 25 or 50 years, depending on the building type, and yet no RSLV have been established, or SLPs made based on experimental work. The way forward to SLPs that account for climate change is thus clear: establish reference values based on accelerated aging methods that integrate climatic conditions expected in 25 to 50 years. This type of work will require significant computer-modelling efforts, both in terms of climatic models, which is well underway, and material degradation models, which must be able to integrate all the degradation factors, including UV radiation, moisture, temperature conditions, mechanical stresses and biological factors.

**Author Contributions:** Conceptualization, M.R.; investigation, M.H.; resources, M.H.; writing—original draft preparation, M.H.; writing—review and editing, M.H., M.R., J.-F.M.; supervision, M.R.; project administration, M.R.; funding acquisition, M.R. All authors have read and agreed to the published version of the manuscript.

**Funding:** This research received no external funding.

**Institutional Review Board Statement:** Not applicable.

**Informed Consent Statement:** Not applicable.

**Conflicts of Interest:** The authors declare no conflict of interest.

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
