# Peer review of "Critical Review of Polymeric Building Envelope Materials: Degradation, Durability and Service Life Prediction"

_buildings, doi:10.3390/buildings11070299_

Round 1

Reviewer 1 Report

This is a high scientific quality paper. The assumptions adopted are explained in detail. This study describes a detailed and in-depth analysis of the topic in question.

Line 1679-1681 - The authors state that "Climate change also means greater aging due to prolonged periods of higher temperatures, higher levels of UV radiation, and higher RH due to increased wind driven rain" - The authors are discussing the particularities of climate change in Canada. This may not be entirely true in other contexts and for other materials.

I found only minor formatting/language lapses:

Line 120 - please replace "word" by "world"

Table 2 - please remove the last two lines (in pdf, are repeated)

Line 1602 - the reference to Equation is missing

Line 1610 - the reference to Equation is missing

Line 1622 - the reference to Equation is missing

Line 1648 - the reference to Equation is missing

Line 1652 - the reference to Equation is missing

Line 1687 - "service life" instead of "useful life"

Line 1850 - the reference to Equation is missing

Reviewer 2 Report

This paper reported a critical review of the degradation, durability and service life prediction of polymer building envelope materials. The paper has been written well. However, certain concerns need to be noticed and clarified:

  1. Lines of 78 and 342, missing space at the beginning of paragraph according to other paragraph format.
  2. There are repeated contents in Table 2, such as Factor and Examples, etc.
  3. Pay attention to some grammatical mistakes in the paper, such as line 198.
  4. Please write down the entire name of ASTM in line 240.
  5. The same significant digits are reserved for the thermal conductivity in Table 4.
  6. The letter format in formulas 1-4 should be the same as in the text.

Round 2

Reviewer 2 Report

Can be accepted now.